# Using a composite flow law to model deformation in the NEEM deep ice core, Greenland: Part 1 the role of grain size and grain size distribution on deformation of the upper 2207 meters

Ernst-Jan N. Kuiper[1,2], Ilka Weikusat[2,1,3], Johannes H. P. de Bresser[1], Daniela Jansen[2], Gill M. Pennock[1], Martyn R. Drury[1]

[1]Faculty of Earth Science, Utrecht University, 3508 TA Utrecht, the Netherlands
[2]Alfred Wegener Institute, Helmholtz Centre for Polar and Marine Research, 27570 Bremerhaven, Germany
[3]Department of Geosciences, Eberhard Karls University Tübingen, 72074 Tübingen, Germany

*Correspondence to*: Ilka.Weikusat@awi.de and M.R.Drury@uu.nl

**Abstract.** The effect of grain size on strain rate of ice in the upper 2207 m in the North Greenland Eemian Ice Drilling (NEEM) deep ice core was investigated using a rheological model based on the composite flow law of Goldsby and Kohlstedt (1997, 2001). The grain size was described by both a mean grain size and a grain size distribution, which allowed the strain rate to be calculated using two different model end members: (i) the micro-scale constant stress model where each grain deforms by the same stress and (ii) the micro-scale constant strain rate model where each grain deforms by the same strain rate. The model results predict that grain-size sensitive flow produces almost all of the deformation in the upper 2207 m of the NEEM ice core, while dislocation creep hardly contributes to deformation. The difference in calculated strain rate between the two model end members is relatively small. The predicted strain rate in the fine grained Glacial ice (that is, ice deposited during the Last Glacial Maximum at depths of 1419 to 2207 m) varies strongly within this depth range and, furthermore, is about 4-5 times higher than in the coarser grained Holocene ice (0-1419 m). Two peaks in strain rate are predicted at about 1980 and 2100 m of depth. The prediction that grain-size-sensitive creep is the fastest process is inconsistent with the microstructures in the Holocene age ice, indicating that the rate of dislocation creep is underestimated in the model. The occurrence of recrystallization processes in the polar ice that did not occur in the experiments may account for this discrepancy. The prediction of the composite flow law model is consistent with microstructures in the Glacial ice, suggesting that fine grained layers in the Glacial ice may act as internal preferential sliding zones in the Greenland ice sheet.

## 1 Introduction

Ice sheets regulate global mean sea level (GMSL) by storing large amounts of fresh water in the form of ice on land. As a consequence of increased anthropogenic global warming, the contribution of the Greenland and the Antarctic ice sheet to GMSL rise is likely to increase in the next centuries (IPCC, 2014). It is therefore important to improve the implementation of ice flow in ice sheet models that calculate the discharge of ice into the ocean, since the amount of water stored in ice sheets is

enough to raise GMSL by about 70 m (Alley et al., 2005; Church et al., 2013). The mass balance of an ice sheet depends on the accumulation of snow on the surface, release of meltwater by runoff and the solid discharge via floating ice shelves and calving of icebergs into the ocean (e.g. Petrenko and Withworth, 1999; Marshall, 2006). In the coldest parts of Antarctica, sublimation and wind erosion can be important ablation mechanisms too (Bintanja, 2009). The amount of ice available for

calving and melting depends on the flow of ice from the interior towards the margins of the ice sheet. This flow of ice is controlled by two processes: sliding of the ice over the bedrock, which includes various sub-glacial processes (Zwally et al., 2002; Vaughan and Arthern, 2007; Thoma et al., 2010; Wolovick and Creyts, 2016), and the internal deformation of the polycrystalline ice, which is governed by various processes like dislocation creep, grain boundary migration (strain induced grain boundary migration – SIBM using the terminology of Faria et al., 2014b)) and grain boundary sliding (GBS) (e.g. Duval

et al., 1983; Alley, 1992; Goldsby and Kohlstedt, 1997, 2001; Montagnat and Duval, 2000, 2004; Schulson and Duval, 2009; Faria et al., 2014a).

For large scale flow models, the deformation of the ice polycrystal is approximated in a homogenized way by continuum mechanics principles. Together with balance equations for mass, momentum and energy, continuum mechanics uses the relation between stress and strain rate given by a constitutive relation, also called the 'flow law'. The most commonly

used flow law in ice sheet models is Glen's law (Glen, 1952, 1955; Paterson, 1994), which describes the flow of polycrystalline ice during deformation by a power law relating equivalent strain rate ($\dot{\epsilon}$) to equivalent stress ($\sigma$) according to

$$\dot{\epsilon} \propto \sigma^n, \tag{1}$$

where n=3 for Glen's flow law. Although Glen found different n values during his laboratory experiments (Glen, 1952, 1953, 1955), a value of n=3 is most often used in ice sheet models (e.g. Paterson, 1994). In the following we thus refer to Glen's

flow law always with n=3. In Glen's flow law grain size insensitive (GSI) dislocation creep is assumed to be the dominant deformation mechanism. Variants of this type of strain rate-stress relation with different values for the stress exponent n have been used, ranging from n=1.5 to 4.0, based on experiments at different conditions (Weertman, 1983). However, at the relatively low driving stresses of <0.3 MPa (equal to an equivalent stress of about 0.5 MPa using Equation 12) in terrestrial ice sheets (e.g. Sergienko et al., 2014), Glen's flow law has proved to be inaccurate (e.g. Thorsteinsson et al., 1999; Huybrechts,

2007) and predicts strain rates that are too slow in both the deeper and the fine grained parts of the polar ice sheets. Laboratory experiments (Mellor and Testa, 1969a; Pimienta and Duval, 1987; Duval and Castelnau, 1995; Goldsby and Kohlstedt, 1997, 2001; De La Chapelle et al., 1999; Duval et al., 2000) and flow analysis of ice sheets (Dahl-Jensen and Gundestrup, 1987; Alley, 1992) have shown that under these low stress conditions, ice deformation is best described by a flow law with a stress exponent less than 3.0. However, most polar ice cores are drilled at low stress locations like domes or ridges (e.g. Faria et al.,

2014b), where the ice might deform by different deformation mechanisms, and therefore be described by a different stress exponent than ice along the flanks or the margins of the ice sheet. The exact deformation mechanism of ice at these low driving stresses is still unclear.

Pimienta and Duval (1987) proposed that in the low stress regime with n<3, fine grained ice deformed by glide of dislocations on the basal slip plane accommodated by grain boundary migration. A quantitative model for this mechanism was

developed by Montagnat and Duval (2000). This model is consistent with a stress exponent n =2 and also indicates that deformation of ice at low stress is likely to be grain size sensitive (Montagnat et al., 2003; Duval and Montagnat, 2006).

During experiments with very fine grained ice (with a grain diameter between 3 $\mu$m and 90 $\mu$m), Goldsby and Kohlstedt (1997) found a transition from a GSI creep regime with a power law stress exponent of n=4 at high equivalent stress (σ>3 MPa) to a grain-size-sensitive (GSS) creep regime with n=1.8 and a grain size exponent (p) of 1.4 at medium equivalent stress (1-3 MPa). At low equivalent stresses (<1 MPa) and with the finest grained samples, Goldsby and Kohlstedt found a third creep regime, again grain size independent, but with n=2.4. Recent studies (Saruya et al., 2019) have confirmed the occurrence of GSS creep in fine-grained ice, with n=2 and p=1.4. According to Goldsby and Kohlstedt, the GSI regime with n=4 is governed by dislocation creep using the easy slip systems in ice (basal slip) rate-limited by the hard slip systems (i.e. non-basal slip). In the GSS regime with n=1.8, basal slip is thought to be rate-limited by grain boundary sliding (GBS), while the opposite holds for the low stress GSI regime with n=2.4, where GBS is rate-limited by basal slip. The accommodating mechanisms are required to maintain strain progression and compatibility (e.g. avoid creation of pore space or voids). They are enabling the major mechanism and are the slowest mechanism (Durham and Stern, 2001), and are thus rate-controlling. At even lower stresses a fourth creep regime, diffusion creep, is expected. This creep regime is expected to dominate the flow behaviour of ice at very small grain sizes (<<1 mm), low temperature and very low equivalent stresses. This creep regime was not reached during the experiments of Goldsby and Kohlstedt (2001) and is assumed to be irrelevant for terrestrial ice sheets (Goldsby and Kohlstedt, 2001; Durham et al., 2010). Therefore, diffusion creep will not be considered in the remainder of this paper.

The hypothesis of Goldsby and Kohlstedt (1997; 2001), concerning the role of GBS in ice, has been seriously questioned by Duval and Montagnat (2002), who noted that deformation by dominant GBS is inconsistent with the strong Crystallographic preferred orientations (CPOs) found in polar ice, which can be explained by deformation involving basal slip accommodated by grain boundary migration (Pimenta and Duval, 1987; Montagnat and Duval, 2000). In response, Goldsby and Kohlstedt (2002) noted, that a CPO could develop in the case of GBS accommodated by easy basal slip. While the detailed deformation mechanisms are controversial, there is actually a consensus that the low stress deformation of ice, involves slip on the easy basal slip system accommodated by grain boundary processes (Goldsby and Kohlstedt 2002, Duval and Montagnant (2002) and that deformation in the low stress regime is likely to be grain size sensitive (Montagnant et al, 2003; Duval and Montagnat, 2006).

The recent study by Saruya et al. (2019) confirms the occurrence of GSS creep in fine-grained ice and proposes a different mechanism, where grain boundaries enhance creep by acting as sinks for dislocations. We emphasize that even though the exact mechanisms involved in GSS flow are not understood in detail, there is good experimental evidence for GSS creep in ice and the available flow laws provide a reasonable basis to investigate the role of these mechanisms in polar ice sheets.

Goldsby and Kohlstedt (2001), Goldsby (2006) and Durham et al. (2010) suggested that Glen's flow law actually represents a combination of deformation mechanisms at the stress range 0.1-1.0 MPa (Glen, 1952; 1955), rather than just dislocation creep. This forms a possible explanation for a lack of accuracy in calculating ice strain rates when using Glen's

flow law with fixed n=3 at very low stress (e.g. Peltier et al., 2000; Durham and Stern, 2001; Durham et al., 2010). For instance, Schulson and Duval (2009) claim that the n=4 regime applies to tertiary creep, which was the same stress exponent found in the high pressure experiments (Durham et al., 1983; Kirby et al., 1987). Analysis also indicates that a power law with a stress exponent of n=4 describes best the observed state of the northern part of the Greenland ice sheet (Bons et al., 2018). Other

factors that are often linked to polar ice being softer than predicted by Glen's flow law at low stress are the anisotropy of ice, impurity content and the softening of ice caused by a small grain size (e.g. Fisher and Koerner, 1986; Dahl-Jensen and Gundestrup, 1987; Paterson, 1991; Cuffey et al., 2000a). Attempts have been made to account for these softening factors by introducing a pre-exponential enhancement factor (e.g. Martín et al, 2009; Seddik et al., 2008; Greve, 1997). When adjusted for anisotropy and grain size, this pre-exponential factor can be as high as 20 (Azuma, 1994). Instead of using an enhancement

factor to artificially speed up Glen's flow law, with an assumed value of n=3, a flow law that is directly based on the actual behaviour of ice at steady state or in tertiary creep may be used. Evidence for a stress exponent nearer to 4 has been often been reported (Glen 1953; Peltier et al., 2000; Qi et al., 2017; Treverrow et al., 2012).

The different GSS and GSI creep regimes recognized by Goldsby and Kohlstedt (1997, 2001) have been combined by these authors in the form of one composite flow law. The major advantage of such a composite flow law is that it explicitly

denotes the components of the different creep mechanisms, rather than just describing bulk behaviour. This provides the opportunity to calculate the relative importance of the GSS and GSI deformation mechanisms over the range of temperatures, differential stresses and grain sizes found in ice sheets and compare their contribution to the total strain rate. This is not possible with the flow law developed by Baker (1981) for example, which adds a factor to the pre-existing Glen's flow law to account for the effect of grain size.

It is well known that grain size varies throughout both the Greenland and the Antarctic ice sheets (e.g. Gow et al., 1997; Faria et al., 2014b; Binder, 2014; Fitzpatrick et al., 2014). However, the effect of grain size on strain rate is usually not considered in ice sheet models that apply flow laws of the type of Glen's flow law. Furthermore, the grain size is often expressed as a single mean grain size, which is generally not a good representation for a grain size distribution (Kipfstuhl et al., 2009). For instance, small grains in a distribution might contribute differently to the overall behaviour than large grains,

with a given set of GSS and GSI mechanisms. Ter Heege et al. (2004) showed that using a mean grain size instead of a full grain size distribution in composite flow laws can lead to an over- or underestimate of the strain rates of natural rocks.

In the NEEM ice core in northwest Greenland (77.45°N, 51.06°W, core length 2540 m) (NEEM community members, 2013) and other Greenland ice cores, the dominant deformation mechanism in the Holocene ice is thought to be dislocation creep (e.g. Thorsteinsson et al., 1997; De La Chapelle et al., 1998; Weikusat et al., 2017). Studies of microstructures by light

microscopy suggested that rotation recrystallization was active in the Holocene ice (De La Chapelle et al., 1998), although recent work shows that strain induced grain boundary migration (SIBM) and grain dissection processes are important in maintaining a constant grain size with depth (Faria et al., 2014b; Steinbach et al., 2017). In the Glacial ice (deposited as snow in the Glacial period) the strong crystallographic preferred orientations (CPOs) indicate that large strains are accommodated by basal slip (Thorsteinsson et al., 1997; Montagnat et al., 2014) while subgrain development and clusters of grains with similar

orientation have been interpreted as evidence for rotation recrystallization (Thorsteinsson et al., 1997; De La Chapelle et al., 1998; Obbard et al., 2006). These features suggest that dislocation creep is dominant throughout the entire section of Holocene and Glacial ice in Greenland (e.g. De La Chapelle et al., 1998). In contrast, Goldsby and Kohlstedt (2002) interpreted the occurrence of straight grain boundaries and equant grains in Glacial ice from the GRIP ice core as evidence for GBS, however, this interpretation was not accepted by most members of the ice community (Wilson 2019). More recently, Faria et al. (2014a; and 2014b) noted that if strain is high in impurity-rich, fine-grained cloudy bands in the Glacial ice, flow is likely to be accommodated by GSS deformation, and Saruya et al. (2019) have suggested that ice sheet microstructures with straight grain boundaries could deform by GSS creep with stress exponent of $n = 2$ and grain size exponent $p = 1.4$, while microstructures with irregular grain boundaries and high subgrain density deform by GSI dislocation creep with $n = 3$.

In this paper, the composite flow law is used to investigate the effect of grain size, grain size distribution and different micro-scale models on the dominant deformation mechanism and the predicted strain rate down to 2207 m of depth in the NEEM deep ice core and the results are compared to results obtained with Glen's flow law. The grain size data are described using both a mean grain size and a grain size distribution to evaluate the natural variability in grain size and the effect of grain size variation on predicted strain rate in polar ice. The deformation mechanisms predicted from the composite flow law are compared with the mechanisms inferred from microstructural and CPO studies of the ice cores. The actual variation of strain rate with depth is currently being investigated by continuing bore-hole logging studies (Dahl-Jansen et al., 2016; Dahl-Jansen, pers. comm. 2019) in the NEEM bore hole. Preliminary results (Greve et al., 2017) indicate that enhanced strain rates occur in the NEEM Glacial age layers. This study is in progress and will provide important information on the effect of grain size, CPO and impurity content on ice deformation by providing strain rate measurements with depth to compare with those predicted by the model presented here.

## 2 Methods

### 2.1 Study site and ice microstructure

The NEEM deep ice core was chosen because a comprehensive light microscope data set was available (Kipfstuhl, 2010) enabled by a fast line scan technique with microscopic resolution (LASM - Large Area Scanning Macroscope; Krischke et al., 2015). From these LM images the grain size distribution is available (Binder et al., 2013). The NEEM ice core was drilled between 2008 and 2012 and is located close to an ice ridge (NEEM community members, 2013). The flow at the NEEM ice core is mainly parallel to the ridge with a small divergent component perpendicular to the line of the ridge (Montagnat et al., 2014). The top 1419 m of the NEEM deep ice core consists of ice deposited during the Holocene (Holocene ice), which shows a steadily increasing crystallographic preferred orientation (CPO) towards a vertical c-axis single maximum with a tendency towards a great circle "girdle" distribution (Figure 1a; Eichler et al., 2013; Montagnat et al., 2014). The mean grain area increases from sub-millimetre size at the surface to about 5 mm² at about 400 m of depth, after which it remains approximately constant for the remainder of the Holocene ice (Montagnat et al., 2014). At the transition from the Holocene ice to ice from

the last glacial period (Glacial ice), a sharp decrease in mean grain area to 1-2 mm² is observed and the c-axis vertical clustering is further strengthened. At a depth of 2207 m, which is the transition from the Glacial ice to the ice deposited during the Eemian period, the grain area increases sharply to hundreds of mm² and the shape of the CPO varies strongly with depth (NEEM community members, 2013). Figure 1c shows that the temperature in the Holocene ice is constant at 244K after which the temperature starts to increase in the Glacial and Eemian ice reaching almost pressure melting point close to the bedrock.

The difference in microstructure between the Holocene ice and the Glacial ice of the NEEM ice core is shown in Figure 2. The Holocene ice core section (Figure 2a) was taken from a depth of 921 m and contains coarse grains with an aspect ratio of about 1:1 and has a relatively irregular grain boundary structure. The ice core section from the Glacial ice (Figure 2b) is taken from a depth of 1977 m and is one of the finest grained ice core sections that was used in this study. The LM image shows very fine grained sub-horizontal bands with numerous quadruple junctions. The grains in these fine grained bands are flattened and have an aspect ratio of about 2:1. The fine grained sub-horizontal bands contain many grains with aligned grain boundaries.

In this study, 615 LM images of the Holocene and Glacial ice were used to determine the evolution of the grain size with depth in the NEEM ice core. These LM images were made of sections that were cut parallel to the long (vertical) axis of the ice core (Kipfstuhl et al., 2006). Each LM image is about 90 mm long and about 55 mm wide and was digitally analyzed using the Ice-image software (www.ice-image.org) (Binder et al., 2013; Binder, 2014), which automatically detects the grain area of each grain in the cut sample surface by counting the pixels enclosed by grain boundaries. A lower cut-off grain size diameter of 0.3 mm was used to exclude the small features that are produced as results of sample relaxation around air bubbles and segmentation by the analysis software (Figure 3). Grains with grain boundaries that were interrupted by the edge of the sample, such as the white grain in the upper right corner of Figure 3b, were excluded from the grain size data.

**2.2 The composite flow law**

The composite flow law as proposed by Goldsby and Kohlstedt (2001) is formulated as follows:

$$\dot{\varepsilon} = \dot{\varepsilon}_{disl} + \left( \frac{1}{\dot{\varepsilon}_{basal}} + \frac{1}{\dot{\varepsilon}_{GBS}} \right)^{-1} + \dot{\varepsilon}_{diff}, \qquad (2)$$

where $\dot{\varepsilon}$ is the total strain rate, composed of strain rates for basal slip rate-limited by non-basal slip or dislocation creep, $\dot{\varepsilon}_{disl}$, the strain rate produced by basal slip, $\dot{\varepsilon}_{basal}$, the strain rate produced by GBS, $\dot{\varepsilon}_{GBS}$, and diffusion creep, $\dot{\varepsilon}_{diff}$. Each of these creep mechanisms can be described by a power law relation of the form:

$$\dot{\varepsilon} = A \, \sigma^n \, d^{-p} \, \exp\left(-\frac{Q + PV^*}{RT}\right), \qquad (3)$$

where A is a material parameter, σ is the differential stress (MPa), n is the stress exponent, d is the grain size diameter (m), p is the grain size exponent, Q is the activation energy for the creep mechanism (J mol⁻¹), P is the hydrostatic pressure (MPa), V* the activation volume (m³ mol⁻¹), R is the gas constant (J K⁻¹ mol⁻¹), and T the absolute temperature (K). The effect of PV* is assumed to be very small (Durham and Stern, 2001) and is ignored for the remainder of this study. The power law relationship of Equation (3) also corresponds to the type of flow equation used in Glen's law (Glen, 1955; Paterson, 1994).

As explained by Durham and Stern (2001) and Durham et al. (2001; 2010), GBS and dislocation slip are sequential processes where one cannot proceed without the other and the slowest mechanism determines the overall strain rate, i.e. is rate-limiting, so that if:

$$\dot{\varepsilon}_{basal} \gg \dot{\varepsilon}_{GBS} \text{ then } \left(\frac{1}{\dot{\varepsilon}_{basal}} + \frac{1}{\dot{\varepsilon}_{GBS}}\right)^{-1} \approx \dot{\varepsilon}_{GBS} \qquad (4)$$

$$\dot{\varepsilon}_{basal} \ll \dot{\varepsilon}_{GBS} \text{ then } \left(\frac{1}{\dot{\varepsilon}_{basal}} + \frac{1}{\dot{\varepsilon}_{GBS}}\right)^{-1} \approx \dot{\varepsilon}_{basal} \qquad (5)$$

Under the temperatures, grain sizes and stresses that occur in natural ice on Earth, basal slip will always give faster strain rates than GBS (Equation 4) and non-basal slip (Goldsby and Kohlstedt, 2001; Goldsby, 2006). This makes, according to Goldsby and Kohlstedt (2001), either GBS or non-basal slip the rate-limiting mechanism for deformation of ice sheets. Therefore, basal slip-limited creep is not considered in the remainder of this paper, as this deformation mechanism is not relevant for polar ice sheets. Hence, the composite flow law simplifies to:

$$\dot{\varepsilon} = \dot{\varepsilon}_{disl} + \dot{\varepsilon}_{GBS}. \qquad (6)$$

As noted in the introduction, the mechanisms involved in the low stress, grain size sensitive deformation regime in ice are debated and several different creep models have been proposed (Montagnat and Duval, 2000; Goldsby and Kohlstedt 2001; Saruya et al., 2019). While the deformation mechanisms involved are controversial there is some consensus from the experimental studies on the flow law parameters of this regime (Goldsby and Kohlstedt 2001; Saruya et al; 2019) with n about 2 and p about 1.4. As we are using the simplified version of the composite flow law in equation (6) our study is not dependent on the details of the deformation mechanisms in the GSS regime.

Figure 4 shows the effect of grain size and stress on strain rate for GBS-limited creep, $\dot{\varepsilon}_{GBS}$, the dislocation creep mechanism, $\dot{\varepsilon}_{disl}$, of the composite flow law and the most often used form (n=3) of Glen's flow law. The chosen uniform temperature of 243K is representative for the upper 1200 m at NEEM (Figure 1c; Sheldon et al., 2014). Glen's flow law and the dislocation creep mechanism are not dependent on grain size (p=0), but have a different slope to each other resulting from the different stress exponents. GBS-limited creep shows relatively fast strain rates at small grain sizes and low stresses. At higher stresses and larger grains, the dislocation creep mechanism is dominant over GBS-limited creep.

The smaller grains in a polycrystal can potentially deform by a different mechanism than the larger grains, since the smaller grains are more susceptible to GSS deformation mechanisms. In order to study the effect of the variation in grain size, a grain size distribution was used as well as a mean grain size in the flow law (Freeman and Ferguson, 1986; Heilbronner and Bruhn, 1998; Ter Heege et al., 2004). As shown by Freeman and Ferguson (1986) and Ter Heege et al. (2004), applying a grain size distribution into a composite flow law leads to application of two theoretical end members to describe the deformation of ice: the homogeneous stress, or micro-scale constant stress model, and the homogeneous strain rate, or micro-scale constant strain rate model.

The micro-scale constant stress model assumes that each grain experiences the same stress, which is equal to the bulk stress of the material. However, the strain rate produced by each grain is different. The driving stress and the bulk strain rate produced by this model can be expressed as follows:

$$\sigma_\sigma = \sigma_1 = \sigma_2 = \sigma_i \ , \tag{7}$$

$$\dot{\epsilon}_\sigma = v_1\dot{\epsilon}_1 + v_2\dot{\epsilon}_2 + \ ... \ + v_n\dot{\epsilon}_n = \sum_{i=1}^{n} v_i\dot{\epsilon}_i \ , \tag{8}$$

where the bulk volume has been segmented into n grain size classes, and $v_i$ stands for the volume fraction of grain size class i, and $\dot{\epsilon}_i$ stands for the strain rate of grain size class i.

The micro-scale constant strain rate model assumes that each grain deforms by the same strain rate, which is equal to the strain rate of the bulk material. The stress required to produce this strain rate is different for each grain size class. This end member assumes that the larger grains in the polycrystalline material support more stress than the smaller grains and can be describes as follows:

$$\dot{\epsilon}_{\dot{\epsilon}} = \dot{\epsilon}_1 = \dot{\epsilon}_2 = \dot{\epsilon}_i \ , \tag{9}$$

$$\overline{\sigma}_{\dot{\epsilon}} = v_1\overline{\sigma}_1 + v_2\overline{\sigma}_2 + \ ...+ v_n\overline{\sigma}_n = \sum_{i=1}^{n} v_i\sigma_i \ , \tag{10}$$

where $\overline{\sigma}_i$ stands for the stress supported by grain size class i. An iterative approach is required to calculate the strain rate when the bulk stress is known.

## 2.3 Boundary conditions and input data

The temperature in the NEEM borehole reaches 269.8K at the bedrock interface, which is only 0.9K below the estimated pressure-melting point (Sheldon et al., 2014). At this temperature it is expected that the deformation of ice is affected by premelting (e.g. Mellor and Testa, 1969b; Barnes et al., 1971; Morgan, 1991; Goldsby and Kohlstedt, 2001). To omit the effect of the different temperature thresholds given in Table 1 for Glen's flow law (263K), the GBS-limited mechanism (255K) and the dislocation creep mechanism (258K) of the composite flow law (Goldsby and Kohlstedt, 2001; Goldsby, 2006), the high temperature flow law parameters, and therefore the effect of premelting on strain rate, were not included in this study. Since the effect of premelting is expected to start close to the Glacial-Eemian interface at about T>262K this paper (part 1) focuses on the relatively cold Holocene and Glacial ice in the upper 2207m of NEEM. The deeper possibly premelted ice will be the subject of a companion paper (part 2), which will analyse the combined effects of grain size and premelting on deformation in the NEEM ice core.

The method of Heilbronner and Bruhn (1998) was used to convert 2D sectional areas to 3D volume fractions. This method uses an equivalent grain diameter that was determined for each grain and a 3D volume fraction was calculated with the assumption of a spherical grain. This method corrects for the over representation of small grains in an LM image compared to the bulk volume. The grain size distribution contains 80 grain size classes defined by steps of the equivalent diameter of 0.3 mm each, covering the full breath of the observed grain size distribution in the Holocene and Glacial ice. The equivalent diameter of each grain in each ice core section was calculated and included in the corresponding grain size class. Figure 5

shows an example of the volume contribution of sectional circles and the corresponding volume contribution of spheres for an ice core section at 921 m of depth, (detail of microstructure is shown in Figure 2a).

The mean grain size was determined by dividing the total area, as classified by the "grain" category in the Ice-image software, by the number of grains for each LM image. This way, the areas affected by bubbles, fracturing or frost are excluded from the mean grain size calculation. A mean equivalent diameter for each ice core section was calculated by assuming a circular grain. Since the composite flow law of Goldsby and Kohlstedt (1997, 2001) requires a grain diameter instead of grain area as an input variable, the conversion of grain area to equivalent grain diameter is required. This conversion might induce a small but systematic error, since recrystallized materials tend to have a log normal grain size distribution, which leads to the calculated equivalent grain diameter to be larger than the mean grain diameter. When using a mean grain size, there are no series of volume fractions (Equation 8 and 10) so no application of the micro-scale constant stress model and the micro-scale constant strain rate model is required.

To calculate the strain rate using Glen's flow law and the composite flow law at the location of the NEEM ice core, information about the variation of stress with depth in the ice sheet is required. As stress itself cannot be measured, the stress has to be estimated based on theoretical considerations and constraints on strain rates in the NEEM ice core.

The shear stress in an ice sheet is driven by gravity and is determined by the surface slope of the ice sheet and the depth from the surface $z_{ice}$. The shear stress along an ice core can be estimated using the shallow ice approximation (e.g. Hutter, 1983; Greve and Blattter, 2009):

$$\tau = -\rho_{ice} \, z_{ice} \, g \, \frac{\partial h}{\partial x}, \qquad\qquad (11)$$

where τ is the shear stress (Pa), $\rho_{ice}$ is the density of ice (910 kg m³), $z_{ice}$ is the ice thickness (m), g is the gravitational constant (9.81 m s²) and $\frac{\partial h}{\partial x}$ is the surface slope in the direction of flow. The surface slope at NEEM is about 1.8 m km¹ (Montagnat et al., 2014) and the ice core length is 2540 m (Rasmussen et al., 2013). Assuming a constant ice density, the shear stress increases linearly with depth reaching 0.041 MPa at the ice-bedrock interface at NEEM. Both the composite flow law and Glen's flow law were derived during uniaxial deformation tests in secondary creep (Glen, 1952, 1955; Goldsby and Kohlstedt, 1997, 2001). So in order to use the shear stress as input for the flow laws, the shear stress has to be converted to an equivalent stress σₑ using the following relationship (Paterson and Olgaard, 2000):

$$\sigma_e = \sqrt{3} \, \tau, \qquad\qquad (12)$$

which results in an equivalent axial differential stress at the ice-bedrock interface at NEEM of 0.071 MPa as shown in Figure 6.

The upper part of the NEEM ice core is dominated by longitudinal stress, perpendicular to the plane of the divide, leading to thinning of the annual layers (Dansgaard and Johnsen, 1969; Montagnat et al., 2014). Longitudinal stresses can be calculated from the increase of the ice slope away from the divide, if the rheology of ice is assumed (Raymond, 1983; Dahl-Jensen, 1989a). In this study, the simple approach of assuming an imposed stress-depth relationship will be taken to investigate

how ice rheology is influenced by grain size, temperature and imposed stress. The layer thinning in the upper part of an ice core, caused by the increased overburden pressure resulting from the deposition of a new snow layer each year, provides a constraint on the vertical strain rate. In case of the NEEM ice core this gives a value of about $3.2 \ 10^{-12} \ s^{-1}$ (Gillet-Chaulet et al., 2011; Montagnat et al., 2014) in the Holocene ice. This strain rate can be used to estimate the equivalent stress in the

upper part of the ice sheet using the composite flow law and Glen's flow law (Figure 6). A constant equivalent stress value of 0.07 MPa using the composite flow law, reproduced the rate of observed layer thinning, as shown in Figure 9. (Note that Figure 9 is based on the composite flow law with the modified flow law parameters, discussed in Section 3). For Glen's flow law the equivalent stress required to reproduce the observed layer thinning in the Holocene ice is lower at about 0.04 MPa. It was therefore decided to assume a constant equivalent stress of 0.07 MPa along the length of the NEEM ice core as input for Glen's

flow law and the composite flow law. At the base of the ice sheet the vertical equivalent stress will tend to zero, with the decrease of vertical stress depending on the stress exponent in the flow law (Dansgaard and Johnson, 1969; Dahl-Jensen, 1989b).

      The assumption of constant equivalent stress with depth is not realistic for ice sheets in general (e.g. Dahl-Jensen, 1989b). However, this assumption is a useful first approximation for the NEEM ice core where the equivalent stress, related

to the shear stress in the lower part of the ice core, is by coincidence similar to the magnitude of equivalent stress related to the vertical stress in the upper part of the ice core. While this approach is simple, it is a useful first step. We have also investigated the effect of changing the constant equivalent stress in the range of 0.01 to 0.50 MPa. When calculating strain rates, no distinction is made between simple shear and vertical flattening deformation.

**3 Flow law parameters**

The most recently updated flow law parameters for the simplified composite flow law (Goldsby, 2006) and the most often used form of Glen's flow law with n=3 (Paterson, 1994) are given in Table 1 for low temperatures where no premelting effects are expected. In order to apply the composite flow law of Goldsby and Kohlstedt (2001) to the NEEM ice core, a check was made of the various published flow laws (Goldsby and Kohlstedt, 1997, 2001; Goldsby, 2006) with the experimental data.

This analysis highlighted an error in one part of the published flow law. The error was confirmed by others (D. Goldsby, personal communication 2018, D. Prior, personal communication 2018) and is explained and corrected in the next two paragraphs.

      A comparison of the calculated strain rate for dislocation creep with the experimental data points from Figure 7 of Goldsby and Kohlstedt (2001) was made. The calculated strain rates were based on the flow law parameters in Table 1 and

using Equation (3). This comparison is shown in Figure 7a. The solid blue line shows the calculated strain rate for dislocation creep when the flow law parameters of Table 1 were used and forced with a stress of 6.3 MPa, which is the same stress as used in Figure 6 of Goldsby and Kohlstedt (2001). The calculated strain rate does not coincide with the three experimental data

points for a temperature of <258K. The calculated strain rate is about 15 to 20 times higher than the experimental strain rates. An Arrhenius plot for GBS-limited creep (Figure 7b) was also calculated using a stress of 0.53 MPa and a uniform grain size of 73 $\mu$m, similar to Figure 4 of Goldsby and Kohlstedt (2001) and using the flow law parameters of Table 1. For this deformation mechanism, the calculated strain rate agrees well with the experimental data points at T<255K of Goldsby and

Kohlstedt (2001).

We have modified the flow law parameters for dislocation creep (dashed blue line, Figure 7a). A transition temperature of 262K was taken for both dislocation creep and GBS-limited creep as this is the expected temperature threshold at which premelting starts to dominate ice rheology (part 2 of companion paper). The modified flow law parameters for dislocation creep that are proposed here are shown in Table 2; the flow law parameters for GBS-limited creep are the same as

given in Table 1 except for the temperature threshold. For dislocation creep, the material parameter A and the activation energy Q (Table 2) change significantly compared to the values given by Goldsby and Kohlstedt (2001) and Goldsby (2006) shown in Table 1. These modified flow law parameters show a better agreement with the experimental data points for dislocation creep (Figure 7a) and results in dislocation creep being 15 to 20 times slower compared to the original flow law parameters. We will only show and discuss the results obtained using the flow law parameters given in Table 2.

For both the composite flow law and Glen's flow law, the influence of CPO on strain rate is not taken into account during this study, for example by a pre-exponential enhancement factor. It is well known that dislocation creep in ice is strongly influenced by CPO development (Azuma 1994) and it is likely that flow involving basal slip rate-limited by GBS is also strongly influenced by CPO. The effect of CPO development on the grain size sensitive mechanisms cannot be included yet as experimental data are lacking.

As noted in the introduction, we did not include any grain size evolution in the model, so the model simply applies to the current grain sizes found in the NEEM ice core. It is well known that grain size and grain size distributions are not fixed parameters in ice depending on the recrystallization mechanisms and on parameters such as stress, strain and temperature (Wilson et al., 2014; Faria et al. 2014b; Peternell et al., 2019). The grain size produced by dynamic recrystallization will scale with the stress (Jacka and Li Jun, 1994). As larger grain sizes are produced at lower stress, dynamic recrystallization can limit

the importance of GSS creep (de Bresser et al., 2001). The grain sizes in the GRIP ice core from Greenland and the Byrd ice core from Antarctic are smaller than the stress-grain size relationship and the GSI-GSS transitions (Goldsby and Kohlstedt, 2002, figure 2; Goldsby, 2006 figure 60.5) indicating that GSS creep is potentially favoured in the upper sections of polar ice sheets, compared to the coarser grained deeper ice.

## 4 Results

Figure 8a and 8b show the strain rate per grain size class and the contribution of the two deformation mechanisms (Equation 6) for an ice core section at 921 m of depth (see detail in Figure 2a). This ice core section is located in the middle of the Holocene ice and has a relatively high variation in grain size. Since dislocation creep is a GSI mechanism, the strain rate

produced by this deformation mechanism is the same for each grain size class in the micro-scale constant stress model (Figure 8a). GBS-limited creep shows faster strain rates than dislocation creep for all grain size classes, and strongly decreases in strain rate with increasing grain size, which is consistent with the inverse relationship to grain size given in Equation (3). The volume contribution of each grain size class (black bars in Figure 8a and 8b) is used in Equation (8) to calculate the bulk strain rate for this ice core section and in Equation (10) to iteratively calculate the stress supported by each grain size class.

The total strain rate produced by each grain size class is set to be the same for the micro-scale constant strain rate model (Figure 8b). The relative contribution of each deformation mechanism differs between grain size classes. GBS-limited creep is the dominant deformation mechanism for the smallest grains, whereas dislocation creep becomes increasingly more important for classes with larger grain sizes. However, even for the largest grains in this ice core section the strain rate produced by GBS-limited creep is still slightly larger than the strain rate produced by dislocation creep.

The stress supported per grain size class for the micro-scale constant stress and micro-scale constant strain rate model for the composite flow law is shown in Figure 8c. The stress supported per grain size class using the micro-scale constant strain rate model is used in Equation (10) to iteratively calculate the stress in the bulk material of the ice core section. The smallest grain size classes support only a small amount of stress, since they are more sensitive to GBS-limited creep. As a consequence, the larger grains support more stress and activate a significant amount of dislocation creep.

Figure 9 shows plots of the equivalent strain rate as a function of depth for Glen's flow law and the composite flow law with its two different deformation mechanisms and the three different model end members calculated using the modified flow law parameters for dislocation creep (Table 2) and the original parameters for GBS-limited creep (Table 1). The contribution of GBS-limited creep to bulk strain rate is also shown. The temperature input for all the models is shown in Figure 1c. The results using the full grain size distribution with the micro-scale constant stress model (Figure 9a) and the micro-scale constant strain rate model with the grain size distribution (Figure 9b) are shown, as well as the results using the mean grain size model (Figure 9c). All models show similarities, such as (i) a relatively constant strain rate between 400 m and 1400 m of depth, (ii) a strain rate increase below 1400 m of depth and (iii) a higher strain rate for Glen's flow law compared to the composite flow law along the entire depth range down to 2207 m of the NEEM ice core. In all of the three model end members, dislocation creep hardly contributes to the overall strain. The calculated strain rate for GBS-limited creep and the composite flow law both show a more variable strain rate below 1400 m. This depth coincides with the transition from the coarse grained Holocene ice to the finer grained Glacial ice and with an increase in temperature (Figure 1c). Two strain rate peaks occur at about 1980 m and 2100 m of depth and show a two to three-fold increase in strain rate compared to the mean strain rate in the Glacial ice predicted by the composite flow law.

The difference between the strain rates of GBS-limited creep and dislocation creep in the Holocene ice is smaller for the micro-scale constant strain rate model than for the micro-scale constant stress model and the mean grain size model. Compared to the micro-scale constant stress model, the micro-scale constant strain rate model predicts slightly lower absolute strain rates from GBS-limited creep along the entire upper 2207 m of depth of the NEEM ice core (Figure 9a, b) and especially below 1419 m of depth where the strain rate is more variable. Using a mean grain size produces a weaker variability in strain

rate with depth than the two model end members using a grain size distribution (Figure 9c). The average strain rate in the Holocene ice is about 60% higher using the micro-scale constant stress model compared to the micro-scale constant strain rate model. For the Glacial ice, this difference between the two model end members is about 40%.

To study the dominant deformation mechanism of the composite flow law at different stress levels as well as its
comparison in strain rate to Glen's flow law, both flow laws were forced at different stress values, which roughly cover the range of equivalent stresses in the Greenland and Antarctic ice sheets (Sergienko et al., 2014). Figure 10a shows the calculated strain rate for dislocation creep and GBS-limited creep using the grain size distribution with the micro-scale constant stress model and the temperature profile of NEEM, forced with different constant stress values. At the lowest stress of 0.01 MPa, the strain rate produced by dislocation creep is about three orders of magnitude lower than the strain rate produced by GBS-limited
creep. With increasing stress, the contribution of dislocation creep to the overall strain rate becomes bigger and at a stress of 0.25 MPa, dislocation creep and GBS-limited creep have roughly the same contribution. The strain rate produced by the dislocation creep mechanism at 0.50 MPa is roughly five times higher than for GBS-limited creep. A similar graph showing the calculated strain rate for the composite flow law and Glen's flow law is shown in Figure 10b. At the lowest stress of 0.01 MPa the composite flow law predicts a slightly higher strain rate compared to Glen's flow law. However, with increasing
stress, Glen's flow law predicts progressively higher strain rates than the composite flow law. At the highest stress of 0.50 MPa, the strain rate predicted by Glen's flow law is almost an order of magnitude higher than the strain rate predicted by the composite flow law.

## 5 Discussion

We will first discuss the results from the different micro-scale end member models for distributed grain sizes in section 5.1,
then compare the results of models with a distributed grain size against a mean grain size in section 5.2. The effect of stress levels on predicted strain rates will be discussed in section 5.3 followed by a consideration of the variation of predicted strain rates with depth in section 5.4. In the last section, the predictions of the composite flow law are compared with the deformation mechanisms indicated by the microstructures and CPOs from the NEEM ice core.

### 5.1 Comparison of micro-scale constant stress versus micro-scale constant strain rate model

The main difference between the micro-scale constant stress model and the micro-scale constant strain rate model is that the micro-scale constant stress model allows the smallest grains to deform more than an order of magnitude faster than the larger grains (Figure 8a), while this is not possible in the micro-scale constant strain rate model (Figure 8b). For the micro-scale
constant strain rate model, the strain rate is set to be the same for each grain size class. GBS-limited creep and dislocation creep are therefore co-dependent, since the sum of the two deformation mechanisms has to add up to a certain strain rate

(Equation 6). Consequently, since the strain rate produced by GBS-limited creep decreases with increasing grain size, the contribution of dislocation creep to bulk strain rate increases with grain size. This effect is shown in Figure 8b where the bulk strain rate is similar for each grain size class, but the contribution of dislocation creep to the bulk strain rate increases with increasing grain size. Due to this co-dependence of dislocation creep and GBS-limited creep in the micro-scale constant strain rate model, the strain rate produced by dislocation creep varies slightly with depth as is shown in Figure 9b.

For most ice core samples, the finest grain size classes contribute only slightly to the bulk volume of the material, as is also the case for the ice core section at 921 m of depth shown in Figure 8. For this particular ice core section, the smallest grain size classes (<0.9 mm) make up only 2.7% of the bulk volume. Therefore, the contribution of these smallest grain size classes to the bulk strain rate remains limited. Nevertheless, the average strain rate in the Holocene ice calculated using the micro-scale constant stress model is 60% higher than the average strain rate produced by the micro-scale constant strain rate model. In the Glacial ice, where the grain sizes are finer and grain size distributions are more uniform, the difference in strain rates between the micro-scale constant stress and micro-scale constant strain rate model is 40%. These differences are remarkably small compared to results obtained for the two model end members in wet olivine (Ter Heege et al., 2004) and calcite mylonites (Herwegh et al., 2005). In wet olivine, the bulk strain rate could be up to an order of magnitude higher for the micro-scale constant stress model compared to the micro-scale constant strain rate model for samples with a high standard deviation in grain size distribution. This indicates that the grain size variation measured within each of the 90 x 55 mm ice core sections is not large enough to change the strain rate by an order of magnitude as observed for wet olivine.

By assuming that each grain deforms by the same amount in the micro-scale constant strain rate model, the strain heterogeneities that have often been observed in ice core microstructures (e.g. Obbard et al., 2006; Weikusat et al., 2009a; Faria et al., 2014a; Piazolo et al., 2015; Jansen et al., 2016) are ignored. Contrary to the micro-scale constant stress model, where the contribution of the finest grains is relatively large compared to their volume contribution, the micro-scale constant strain rate model probably overestimates the role of the larger grains on the bulk strain rate in the ice core section. Therefore, the two models represent the lower and upper limit of deformation behaviour of a polycrystal with a distributed grain size (Ter Heege et al., 2004). Natural deformation in ice sheets is likely to involve micro-scale variations of stress and strain rate (e,g, Faria et al., 2014a; Piazolo et al., 2015).

## 5.2 Comparing grain size distribution model end members with the mean grain size model

The difference in calculated strain rate between using a grain size distribution with the micro-scale constant strain rate model and the mean grain size model is relatively small (Figure 9b and c). A single mean grain size eliminates the effect that smaller grains have on the bulk strain rate. However, the effect of the smaller grains on strain rate in the micro-scale constant strain rate model is also limited since all grain size classes deform by the same amount and thus the difference in bulk strain rate between the micro-scale constant strain rate model and mean grain size model is small. The much larger computational expense of the micro-scale constant strain rate model and the small difference in calculated strain with the mean grain size model argues

for using a mean grain size model over a micro-scale constant strain rate model when modelling GSS behaviour in polar ice sheets.

The difference in calculated strain rate between the micro-scale constant stress model and using a mean grain size model is larger than the difference between the micro-scale constant strain rate model and the mean grain size model. The strain rate peaks predicted in the layers at about 1980 m and 2100 m of depth are two to three times larger for the micro-scale constant stress model compared to the mean grain size model (Figure 9a, c). This difference is mainly caused by the effect that the finest grains have on the bulk strain rate in the micro-scale constant stress model (Figure 8a).

### 5.3 Effect of stress magnitude on predicted deformation mechansims

Figure 10a shows that, at equivalent stresses below 0.25 MPa, the strain rate produced by GBS-limited creep is higher than the strain rate produced by dislocation creep, while at an equivalent stress of 0.50 MPa the strain rate produced by dislocation creep is higher than GBS-limited creep. The stress sensitivity of dislocation creep is controlled by a stress exponent of n=4, while the stress sensitivity of GBS-limited creep is controlled by n=1.8. Thus, if temperature and grain size remain constant, dislocation creep becomes progressively stronger relative to GBS-limited creep with increasing stress. However, with the temperature profile and grain size data from the upper 2207 m of depth in the NEEM ice core, an equivalent stress of about 0.25 MPa is required for dislocation creep to become as strong as GBS-limited creep. Such a high stress is not reached in the NEEM ice core as the best estimate gives an equivalent stress of 0.07 MPa. However, the driving stress progressively increases from the ice domes and divides towards the margins of the ice sheet, reaching a driving stress of about 0.30 MPa at the margins of the ice sheets (Sergienko et al., 2014). A driving stress of 0.30 MPa corresponds to an equivalent stress of about 0.50 MPa (Equation 12). Therefore, if the temperature and grain size along the NEEM ice core is comparable to the ice along the margins of the ice sheet, the dominant deformation mechanism could switch from GBS-limited creep near domes and divides to dislocation creep near the margins of the ice sheets.

Interestingly, the stress of 0.10-0.50 MPa is within the range of stresses (0.1-1.0 MPa) that was used during the deformation experiments of Glen (1952, 1955). The grain size reported of 1-2 mm by Glen (1952) is similar to the grain size in the Glacial ice of the NEEM ice core. The results in Figure 10a for 0.25 MPa show that the contribution of dislocation creep and GBS-limited creep to the total strain rate in the Glacial ice is roughly equal. This result supports the hypothesis of Durham et al. (2001, 2010), Goldsby and Kohlstedt (2001, 2002), Goldsby (2006) and Bons et al. (2018) that the stress exponent of n=3 found by Glen (1952, 1955) is the result of collecting data at a transition regime between n=4 for dislocation creep and n=1.8 for GBS-limited creep. Empirical evidence for a change in stress exponent in ice sheets was found by Pettit and Waddington (2003). These authors showed that a low stress exponent is best for describing ice deformation at low stress, while a higher stress exponent is required in high stress environments.

Comparison of the results using the composite flow law with Glen's flow law at different equivalent stresses (Figure 10b) shows that at a stress of 0.01 MPa the composite flow law predicts a higher strain rate than Glen's flow law in most of the upper 2207 m of depth in the NEEM ice core. In the finest grained parts of the Glacial ice, the composite flow law predicts

a strain rate that is about five times faster than predicted by Glen's flow law at the low equivalent stress of 0.01 MPa. As the equivalent stress increases, Glen's flow law becomes progressively faster relative to the composite flow law. Since the dominant deformation mechanism of the composite flow law at low equivalent stress is GBS-limited creep (Figure 10a), the effective stress exponent will also be close to the stress exponent for GBS-limited creep (n=1.8). Glen's flow law, driven by a stress exponent of n=3, shows therefore a stronger increase in strain rate with increasing stress than the composite flow law. Consequently, at a driving stress of 0.25 MPa, Glen's flow law predicts a strain rate that is about an order of magnitude faster than the strain rate predicted by the composite flow law. As the contribution of dislocation creep in the composite flow law increases with increasing stress (Figure 10a), the sensitivity of the bulk strain rate to grain size variation decreases with increasing stress since dislocation creep is insensitive to grain size. This effect can be seen in Figure 10b where the strain rate peaks and the layer-to-layer variability predicted by the composite flow law become weaker with increasing stress.

## 5.4 Variability of predicted strain rates with depth

Levels of high borehole closure and borehole tilting have been observed in many polar ice cores and often coincide with high impurity content and small grain sizes (e.g. Fisher and Koerner, 1986; Paterson, 1994). These depth levels can be seen as layers with a different microstructure than the surrounding ice and therefore deform at a higher strain rate. The reason that small grain size and high impurity content coincide is still not well understood (Eichler et al., 2017). The results using the composite flow law suggest that prominent soft layers, i.e. layers with a high strain rate, could be present at two depths of about 1980 and 2100 m in the NEEM ice core. The predictions from the composite flow law are consistent with preliminary results from ongoing borehole tilt measurements at NEEM (Dahl-Jansen et al., 2016, Greve et al., 2017; Dahl-Jensen pers. comm. 2019). These soft layers are located in the lower part of the NEEM ice core, which is dominated by simple shear (Dansgaard and Johnson, 1969; Montagnat et al., 2014). The soft layers can therefore be seen as depths where a high rate of simple shear occurs, instead of layers with enhanced extrusion (Waddington, 2010). It is likely that not all soft layers that are caused by finer grains have been identified in the model since the available sampling rate of 615 LM images along 2207 m of depth of the NEEM ice core leaves many depth intervals not studied. Glen's flow law is unable to predict soft layers related to grain size variations since the flow law is forced by stress and temperature only. The effects of anisotropy, grain size and/or impurity content on strain rate are often incorporated in the form of an enhancement factor (Azuma, 1994; Thorsteinsson et al., 1999). However, information about the softening effects, like grain size, are needed in order to incorporate the enhancement factor with the right value and at the right depth. This can only be achieved by a flow law that explicitly describes GSS deformation, such as the flow laws of Goldsby and Kohlstedt (2001) or Saruya et al. (2019). GSS deformation is also expected for flow involving basal slip accommodated by grain boundary migration (Montagnant et al. 2003), although there is no calibrated quantitative flow law for this mechanism.

Another reason why soft layers might have been missed during this study is that by taking the grain size distribution of the 90 x 55 mm LM images, the fine grain size of shear bands, kink bands (tilted lattice bands), or cloudy bands (Faria et al., 2010; 2014a, Jansen et al., 2016) could have been averaged out. Often, these bands have a vertical thickness that is much

thinner than the 90 mm height of the LM images (see examples in Figure 2b; Figure 4 in Faria et al., 2014b). Therefore, it is likely that many soft layers in the Glacial ice of the NEEM ice core have not been identified in this study or have been averaged out over the 90 x 55 mm LM images.

**5.5 Comparison of model predictions with deformation mechanisms inferred from NEEM microstructures and CPO.**

The microstructures and CPOs in the NEEM core provide constraints on the active deformation mechanisms. The microstructures in the Holocene ice (Figure 2a) with irregular grain boundaries indicating strain-induced grain boundary migration (SIBM using the terminology of Faria et al., 2014b), and subgrain development in the grains indicate that dislocation creep and dynamic recrystallization are important (Montagnat et al., 2014; Weikusat et al., 2017, Steinbach et al,. 2017). The microstructures in the Holocene ice are clearly different from the microstructure developed in experimental samples by GSS

creep (Goldsby and Kohlstedt, 1997; Saruya et al., 2019). So the results from the composite flow model (Figure 9) are inconsistent with the microstructures in the Holocene ice.

In the Glacial ice different microstructures occur and the grain size is smaller than in the Holocene ice. In the Glacial ice grain boundaries are straight to smooth and the grain aspect ratio found in sub-horizontal fine grained bands (Figure 2b) is similar that reported by Goldsby and Kohlstedt (1997, Figure 6 therein) in the GBS-limited creep experiments. Also, similar

to Goldsby and Kohlstedt (1997), aligned grain boundaries and numerous quadruple junctions were found (Figure 2b). These flattened grains, aligned grain boundaries and quadruple junctions were only found in the finest grained parts of the Glacial ice, which could well indicate that these layers have deformed by GBS, possibly enforced by microshear (Faria et al 2014b).

The c-axis eigenvalues show a minor variability in the Glacial ice of the NEEM ice core (Eichler et al., 2013; Montagnat et al., 2014) where the layers of high GSS creep strain rate are predicted. The strong development of CPO and the

development of substructures indicate that large amounts of strain are accommodated by basal slip of dislocations in the NEEM ice core. This may seem to be in disagreement with the conclusion that GSS creep is the dominant deformation mechanism. A random or weak CPO is usually associated with dominant GBS deformation. However, the paradigm that the presence of a strong CPO rules out significant GBS, is based on studies of isotropic metals and may not always apply to anisotropic materials like minerals and ice (Hansen et al., 2011, 2012).  In their original paper Goldsby and Kohlstedt (1997) proposed that GBS

was the dominant strain producing mechanism in the n=2 p=1.4 regime. Later Goldsby and Kohlstedt (2001) and Goldsby (2006) proposed that as basal slip and GBS were sequential processes, the strain would be mainly accommodated by basal slip with GBS as the rate controlling process. Durham and Stern (2001) made the important point that the composite flow equation for sequential mechanisms embedded in equations (2 to 5) is based on the assertion that the sequential mechanisms accommodate the same amount of strain. So if creep in ice does occur by sequential basal slip and GBS then it is likely that a

CPO will develop during high strain deformation. A strong CPO is expected to develop in the case of basal slip accommodated by grain boundary migration (Montagnat and Duval, 2000) and for the deformation mechanism proposed by Saruya et al., (2019) involving dislocation creep enhanced by grain boundaries acting as sinks for dislocations. Ice deforming by the mechanism proposed by Saruya et al. (2019) would be mainly deforming by basal slip so would develop a strong CPO. The

microstructures reported by Saruya et al. (2019) in the grain size sensitive regime are very similar to those found in the fine-grained Glacial ice of the NEEM core (Shigeyama et al., 2019).

For both the composite flow law and Glen's flow law, the influence of CPO on strain rate is not taken into account during this study, for example by a pre-exponential enhancement factor. In bed-parallel simple shear, a strong vertical single maximum CPO produces less strain incompatibilities at grain boundaries by aligning the basal planes of the ice crystals in the direction of the flow. This means that less accommodation of basal slip is required by either non-basal slip or GBS per unit of strain, which causes the strain rate to increase compared to an isotropic ice sample. The strain rate enhancement caused by a well-developed CPO is about 2.3 times stronger in simple shear than in pure shear (Budd and Jacka, 1989; Treverrow et al., 2012). Therefore, the difference in equivalent strain rate between the Holocene and the Glacial ice is probably larger than shown in Figure 9, since the Holocene ice is predominantly deforming by pure shear, while the Glacial ice is predominantly deforming by simple shear (Montagnat et al., 2014). Both deformation mechanisms of the composite flow law assume basal slip to be the dominant strain producing mechanism and being rate-limited by either grain boundary sliding or non-basal slip. We suggest that both dislocation creep and GSS creep are both enhanced by a strong single maximum CPO as a strong alignment of the dominant slip system produces less strain incompatibilities at grain boundaries and triple junctions.

We will now consider the discrepancy between the model predictions and the deformation mechanisms indicated by the NEEM Holocene microstructures. The results from the composite flow law model shown in Figure 9 suggest that ice deformation in the upper 2207 m of depth in the NEEM ice core is almost entirely produced by GSS creep. However, there is evidence that significant non-basal slip is activated in the polar ice sheets (e.g. Weikusat et al., 2009b, b, 2011, 2017). Goldsby (2006) compared the results of the composite flow law using the original flow law parameters (Table 1) to the grain size and strain rate variability with depth in the basal layer of the Meserve glacier, Antarctica (Cuffey et al., 2000b). It was found that the composite flow law overestimates the contribution of GBS-limited creep in the basal part of the Meserve glacier. The in-situ temperature (256K) and the estimated shear stress (0.05 MPa) in the basal layer of the Meserve glacier are fairly similar to the temperature and stress estimated for the Glacial ice in the NEEM ice core (Figure 1c). With the modified flow law parameters for dislocation creep (Table 2), the calculated contribution of GBS-limited creep to total strain rate would be close to 100%, which is similar to the results with the NEEM ice core (Figure 9). Therefore, it is proposed that the composite flow law severely underestimates the strain rate produced by dislocation creep, as high stresses of about 0.25 MPa are required for dislocation creep to become roughly as fast as GBS-limited creep in the NEEM ice core (Figure 10a). This while the large and interlocking grains in the Holocene ice of the NEEM ice core (Figure 2a) argue against GBS as the dominant rate limiting mechanism for basal slip. It is interesting to note that the constant strain rate model predicts a larger role for dislocation creep so this model may be more appropriate for the shallow ice which is deforming mainly by pure shear, with all layers deforming at the same strain rate.

The deformation experiments in the GSI and GSS regime of Goldsby and Kohlstedt (1997, 2001) were performed at low temperature in order to prevent grain growth during the deformation experiments. Significant grain growth during the deformation experiments would have complicated the derivation of the flow law parameters in the GSS creep regime. Goldsby

and Kohlstedt (1997) stated that "grains in deformed samples were equiaxed with straight grain boundaries; irregular grain boundaries typical of dynamical recrystallization by grain boundary migration were not observed". However, suppressing SIBM during the deformation experiments also meant that SIBM could not remove strain incompatibilities at grain boundaries and/or triple junctions. Recrystallization by SIBM is considered an important softening mechanism in polar ice (e.g. Duval et

al., 2000; Montagnat and Duval, 2000; 2004; Wilson et al., 2014). It is well established that SIBM is active at all depths in polar ice cores (e.g. Weikusat et al., 2009a), although the amount of SIBM varies strongly with depth (e.g. Duval and Castelnau, 1995; Kipfstuhl et al., 2009; Faria et al., 2014a). In the NEEM ice core, SIBM is probably less extensive in the finer grained Glacial ice than in the coarser grained Holocene ice, which is supported by the lower grain boundary curvature in the Glacial ice (Binder, 2014). It is therefore proposed that SIBM is an important softening mechanism in the Holocene ice of the NEEM

ice core, while it was suppressed during the deformation experiments of Goldsby and Kohlstedt (1997, 2001). Therefore, the ice during the GSI deformation experiments of Goldsby and Kohlstedt (2001) was probably relatively hard, compared to natural polar ice, as the synthetic ice in experiments was not softened by SIBM. This would have affected the flow law parameters that were derived from the results of these deformation experiments, potentially underestimating dislocation creep strain rates when using the composite flow law.

## 6 Conclusions

In order to study the effect of grain size and grain size variation with depth in polar ice sheets, the composite flow law of Goldsby and Kohlstedt (2001) was used with temperature and grain size data from the upper 2207 m of depth in the NEEM ice core. A constant equivalent stress with depth was assumed, with a magnitude of 0.07 MPa constrained from the surface slope and layer thinning data from NEEM. GSS deformation was described using a mean grain size and a grain size distribution

in combination with two model end members: the micro-scale constant stress model and the micro-scale constant strain rate model. A modification of the flow law parameters for dislocation creep (GSI) in the composite flow law showed a better fit with the experimental data obtained during the deformation experiments of Goldsby and Kohlstedt (1997, 2001).

The difference between the model end members is relatively small with the micro-scale constant stress model predicting higher strain rates than the micro-scale constant strain rate model and using a mean grain size. The results using the

modified flow law parameters and a constant equivalent stress of 0.07 MPa, predict that GSS creep produces almost all deformation in the upper 2207 m of depth in the NEEM ice core. A strain rate increase, mainly resulting from a reduction in grain size, is predicted below 1400 m of depth for all model end members. Two depths in the Glacial ice with a higher strain rate, caused by enhanced GSS creep, are predicted at about 1980 m and 2100 m of depth.

At the grain size and temperature conditions of the NEEM ice core, GSS creep is predicted to be the dominant

deformation mechanism over dislocation creep for equivalent stresses up to about 0.25 MPa. At higher stresses, which occur at the edges of polar ice sheets, dislocation creep is dominant over GSS creep. At low stresses of about 0.01 MPa, the composite flow law predicts a faster strain rate than Glen's flow law. However, the stress exponent of Glen's flow law is higher than the

effective stress exponent for the composite flow law and therefore the strain rate increase with increasing stress is higher for Glen's flow law than for the composite flow law. At NEEM grain size and temperature conditions, Glen's flow law predicts a higher strain rate than the composite flow law at equivalent stresses higher than 0.05 MPa.

The prediction from the composite flow model that GSS creep is the dominant process at all depths is inconsistent with microstructures in the Holocene ice, indicating that the rate of dislocation creep is underestimated in the model. One possible explanation for this is that recrystallization by SIBM was not active during the experiments of Goldsby and Kohlstedt (1997, 2001), while recrystallization by SIBM is an important softening mechanism in the Holocene ice. In the Glacial ice the microstructures and CPO indicate that large strains are accommodated by basal slip, while the occurrence of straight grain boundaries and quadruple points indicate that GBS may be significant. These features are consistent with GSS creep mechanisms involving large activity of the easy slip system, such as basal slip rate-limited by GBS as proposed by Goldsby and Kohlstedt (2001) or grain-size enhanced dislocation creep as recently proposed by Saruya et al. (2019). The composite flow law model and the microstructures in the Glacial ice both indicate that the fine-grained layers in the Glacial ice may potentially act as internal preferential sliding zones in the Greenland ice sheet.

**Acknowledgements**

This work has been carried out as part of the Helmholtz Junior Research group "The effect of deformation mechanisms for ice sheet dynamics" (VH-NG-802). The NEEM light microscope data used in this study has been made available by www.pangaea.de. The authors would like to thank Sepp Kipfstuhl and Tobias Binder for providing data and encouraging discussions. The authors would like to thank all the NEEM Community members who were involved in the preparation of the physical properties samples in the field. This work is a contribution to the NEEM ice core project which is directed and organized by the Center of Ice and Climate at the Niels Bohr Institute and US NSF, Office of Polar Programs. It is supported by funding agencies and institutions in Belgium (FNRS-CFB and FWO), Canada (NRCan/GSC), China (CAS), Denmark (FIST), France (IPEV, CNRS/INSU, CEA and ANR), Germany (AWI), Iceland (RannIs), Japan (NIPR), South Korea (KOPRI), the Netherlands (NWO/ALW), Sweden (VR), Switzerland (SNF), the Unites Kingdom (NERC) and the USA (US NSF, Office of Polar Programs). Thanks to the reviewers, Dave Prior and Chris Wilson and an anonymous reviewer for their detailed reviews and helpful comments.

**Code/Data availability**

The models were calculated using a C++ code and an Excel spreadsheet using the NEEM grain size and temperature data, which are available from Pangaea Data Publisher for Earth & Environmental Science and the flow law parameters used in table 2.

**Author contribution**

EJK prepared the code, the data, the set-up of simulations and conducted model runs. JHPdB developed the algorithm for the code and provided the initial idea. MRD and GMP supervised and initiated the simulation set-ups and interpretations. DJ and IW provided ice core samples and data, glaciological background and data preparation. All authors jointly interpreted results and wrote the manuscript.

**Competing interests**

No competing interests.

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

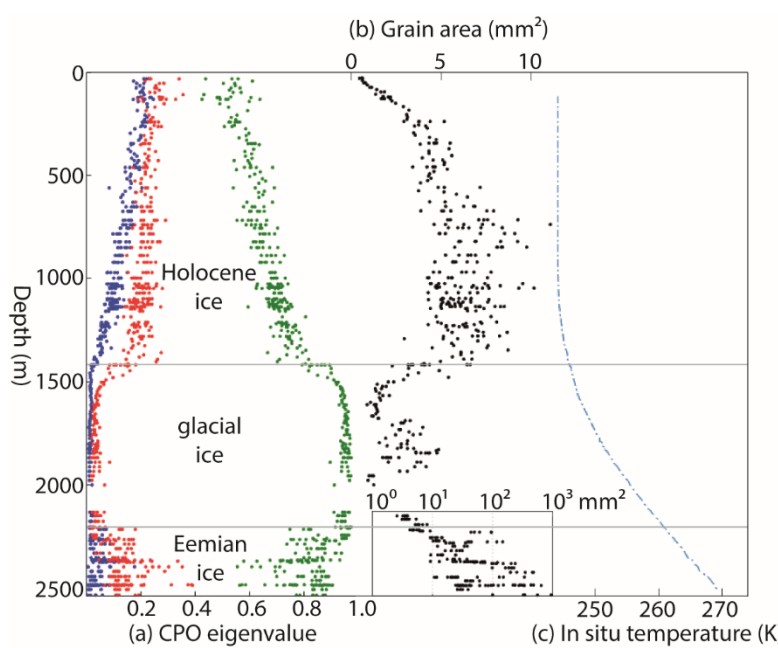

**Figure 1: Compilation of microstructure and borehole data of the NEEM ice core. (a) Orientation tensor eigenvalue from light microscopy studies (blue, red, green) (Eichler et al., 2013). (b) Mean grain area (black dots) (Eichler et al., 2013) and (c) The in-situ temperature (broken blue line) (Sheldon et al., 2014).**

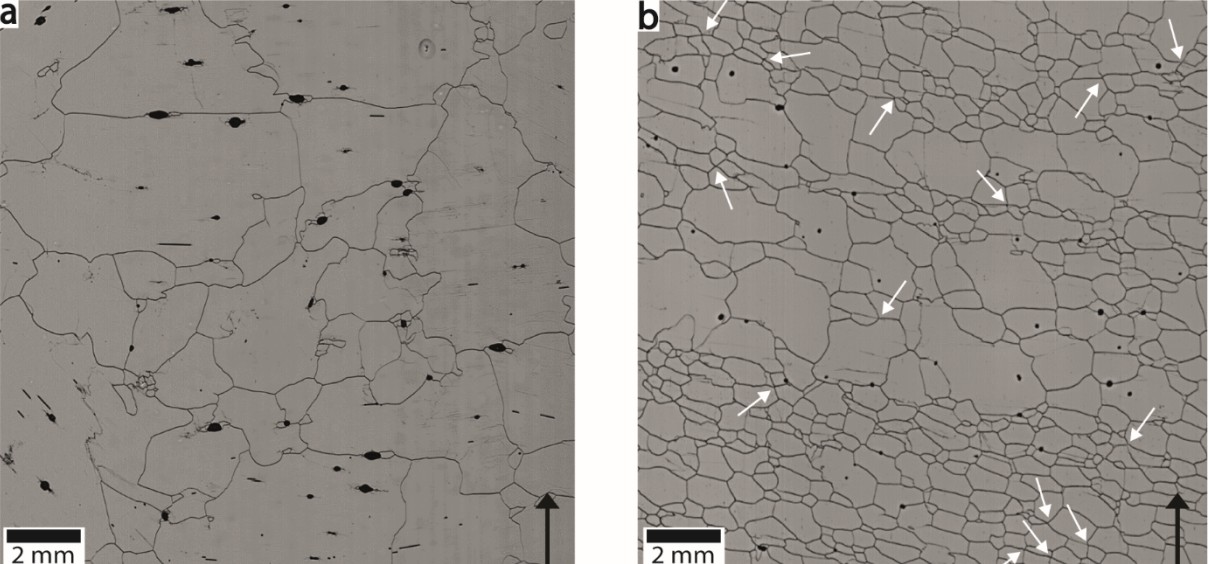

**Figure 2: Reflected LM images of ice core sections at (a) 921 m of depth in the Holocene ice and (b) of 1977 m depth in the Glacial ice of the NEEM ice core. The black arrows indicate the top of the ice core. The Glacial ice core section contains layers with coarse and relatively fine grains that are distributed in sub-horizontal bands. These fine grained sub-horizontal bands have many aligned grain boundaries and quadruple junctions which are indicated by white arrows. Images taken from Kipfstuhl (2010) (doi:10.1594/PANGAEA.743296).**

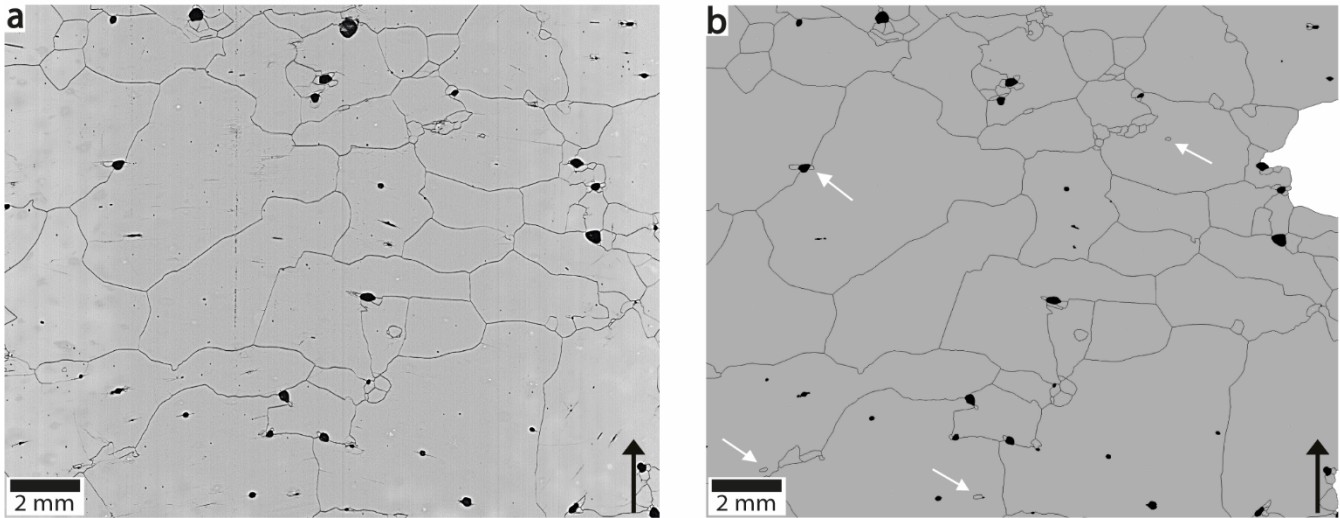

**Figure 3: Reflected LM image (left) and the segmented image (right) of an ice core section at 756 m depth. The black arrows indicate the top of the ice core. White arrows indicate examples of artifacts with a diameter <0.3 mm that were not included in the grain size data. Images taken from Kipfstuhl (2010) (doi:10.1594/PANGAEA.743296) and Binder et al. (2013).**

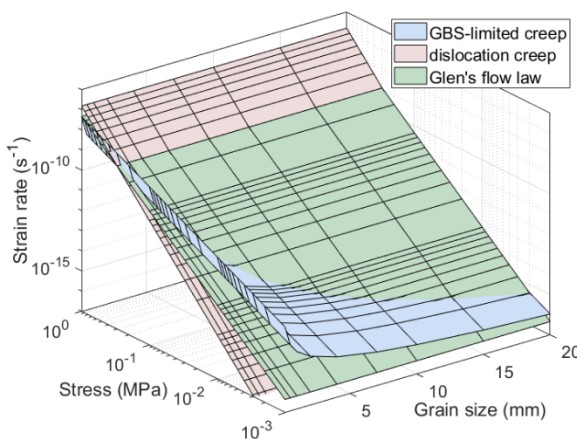

**Figure 4: The effect of grain size and stress on calculated strain rate plotted for Glen's flow, the dislocation creep mechanism and GBS-limited creep of the composite flow law at a constant temperature of 243K using the flow law parameters in Table 1.**

5 **Table 1: Parameters for the simplified composite flow law and for Glen's flow law given by Equation 3 that relates grain size, temperature and stress to strain rate. Values taken from Goldsby and Kohlstedt (2001) and Paterson (1994). The A value for G&K dislocation creep (indicated by \*) was updated and taken from Goldsby (2006).**

| Creep regime | A (units) | n | Q (kJ mol⁻¹) | p |
|---|---|---|---|---|
| G&K dislocation creep (T<258K) | $1.2 \cdot 10^6$ MPa^-4.0 s^-1* | 4.0 | 60 | 0 |
| G&K GBS-limited creep (T<255K) | $3.9 \cdot 10^{-3}$ MPa^-1.8 m^1.4 s^-1 | 1.8 | 49 | 1.4 |
| Glen's flow law (T<263K) | $3.61 \cdot 10^5$ MPa^-3.0 s^-1 | 3.0 | 60 | 0 |

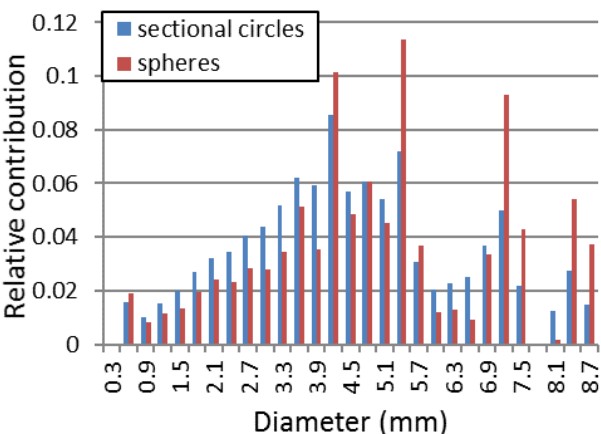

**Figure 5: Relative contribution of each grain size class to the bulk volume of an ice core section containing 965 grains at 921 m depth calculated for sectional circles (blue) and spherical grains (red) using the method of Heilbronner and Bruhn (1988).**

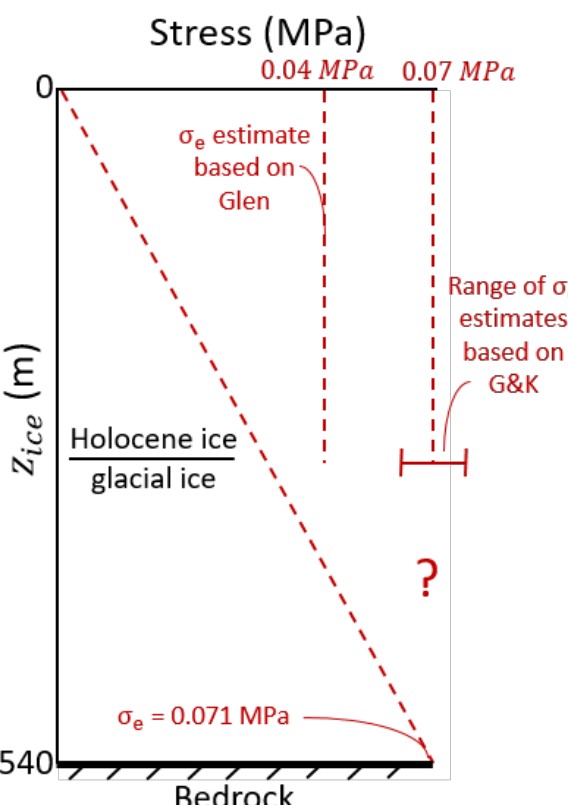

5    **Figure 6: The equivalent shear stress ($\sigma_e$) with depth calculated using the shallow ice approximation (Equation 11) and the range of the estimated longitudinal stress based on the modified parameters of the composite flow law (G&K) of Table 2 and the best estimate for Glen's flow law constrained by the annual layer thinning in the Holocene ice of the NEEM ice core. There are no constraints on the annual layer thinning in the Glacial and Eemian ice, which is indicated by the question mark.**

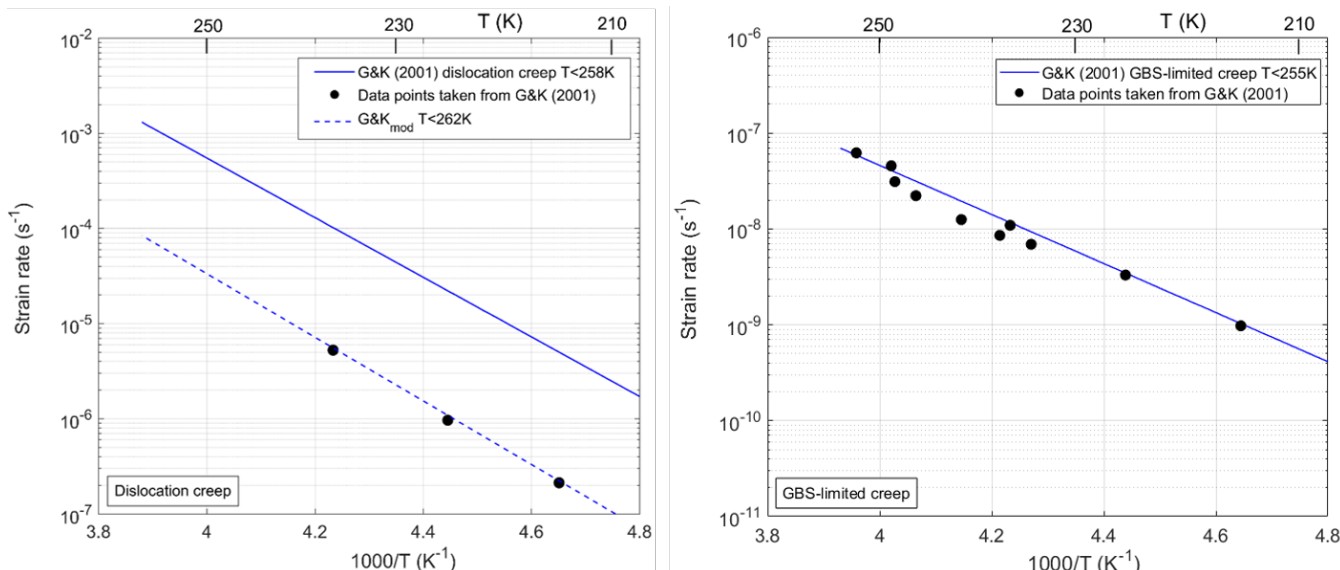

**Figure 7: Arrhenius plot showing (a) the dislocation creep mechanism and (b) the GBS-limited creep mechanism below their temperature thresholds of 258K and 255K, respectively. A stress of 6.3 MPa was used to calculate the strain rates in (a) and a stress of 0.53 MPa and a uniform grain size of 73 $\mu$m was used for (b). The black dots are the experimental data points taken from Goldsby and Kohlstedt (2001). The solid lines represent the calculated strain rate using the original flow law parameters from Goldsby and Kohlstedt (2001) (Table 1). The dotted line, G&K$_{mod}$, is the calculated strain rates for dislocation creep using the modified flow law parameters (Table 2).**

**Table 2: Modified dislocation creep parameters for the composite flow law as derived from Figure 7. The transition temperature is taken to be 262K (see section 2.3 and 3).**

| Creep regime | A (units) | n | Q (kJ mol$^{-1}$) | p |
|---|---|---|---|---|
| G&K$_{mod}$ dislocation creep (T<262K) | 5.0 10^5 MPa^-4.0 s^-1 | 4.0 | 64 | 0 |
| G&K$_{mod}$ GBS-limited creep (T<262K) | 3.9 10^-3 MPa^-1.8 m^1.4 s^-1 | 1.8 | 49 | 1.4 |

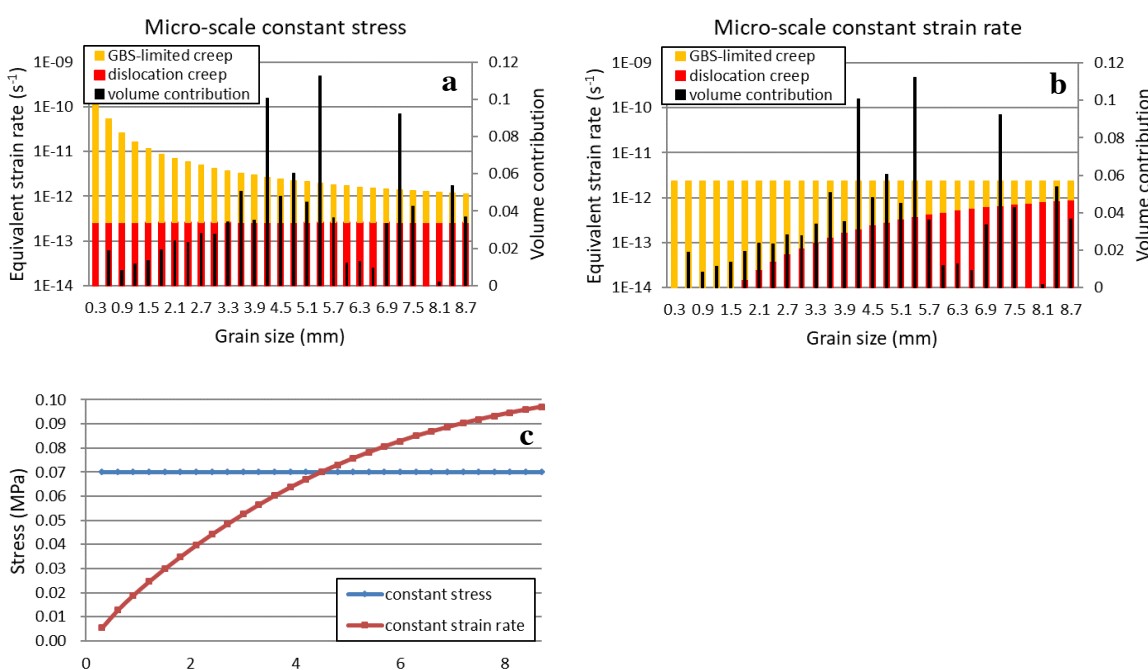

**Figure 8: Grain size class versus log strain rate for an ice core section at 921 m depth for (a) the micro-scale constant stress model, (b) the micro-scale constant strain rate model and (c) the stress supported by each grain size class of the grain size distribution. The results were calculated using the flow law parameters in Table 2. The bulk strain rate for this ice core section using the micro-scale constant stress and micro-scale constant strain rate model following Equations (7) to (10) is $4.1\,10^{-12}s^{-1}$ and $2.4\,10^{-12}s^{-1}$, respectively.**

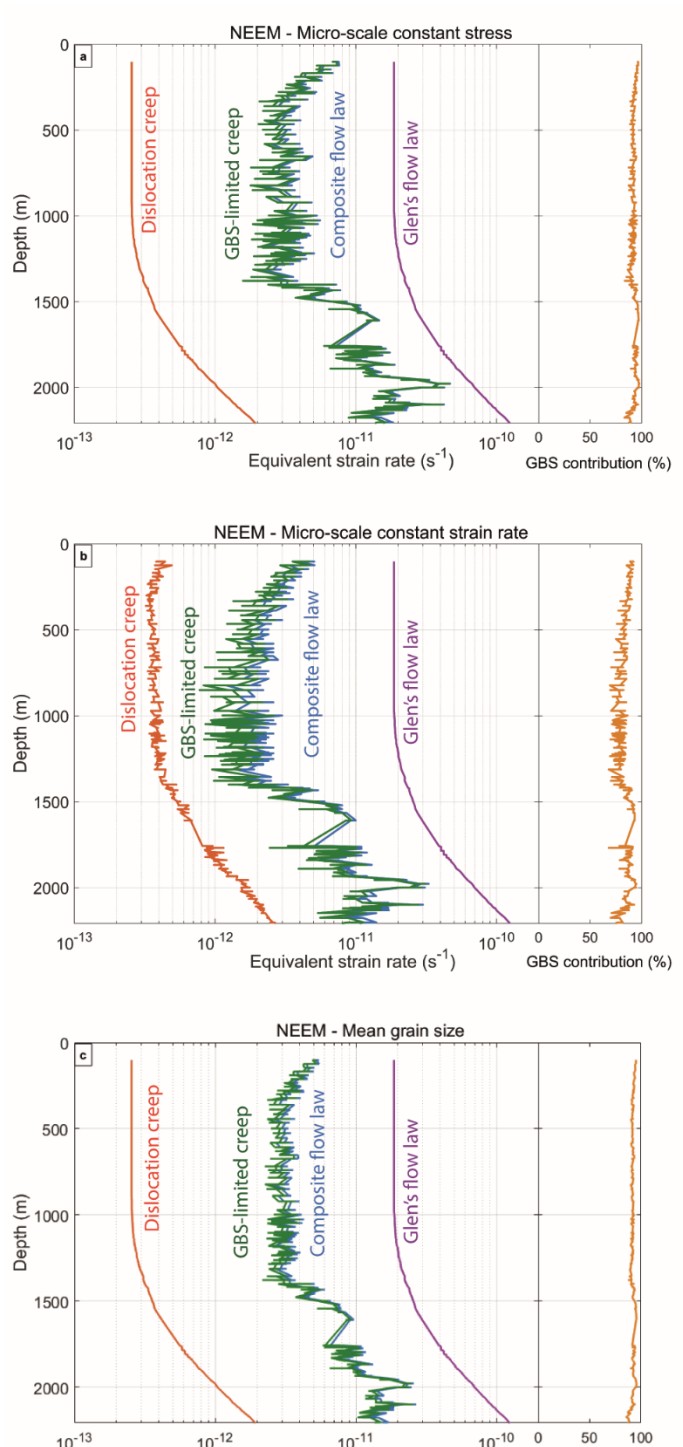

**Figure 9: (a) Results for the Holocene and Glacial ice for Glen's flow law (purple, same in a, b, c) and the composite flow law (blue) using the flow law parameters in Table 2, which consist of dislocation creep (red) and GBS-limited creep (green), using (a) the micro-scale constant stress model with the grain size distribution, (b) the micro-scale constant strain rate model with the grain size distribution and (c) the composite flow law with the average grain size data. The contribution of GBS-limited creep to bulk strain rate for all three model end members is shown. A constant effective stress of 0.07 MPa was used for all figures.**

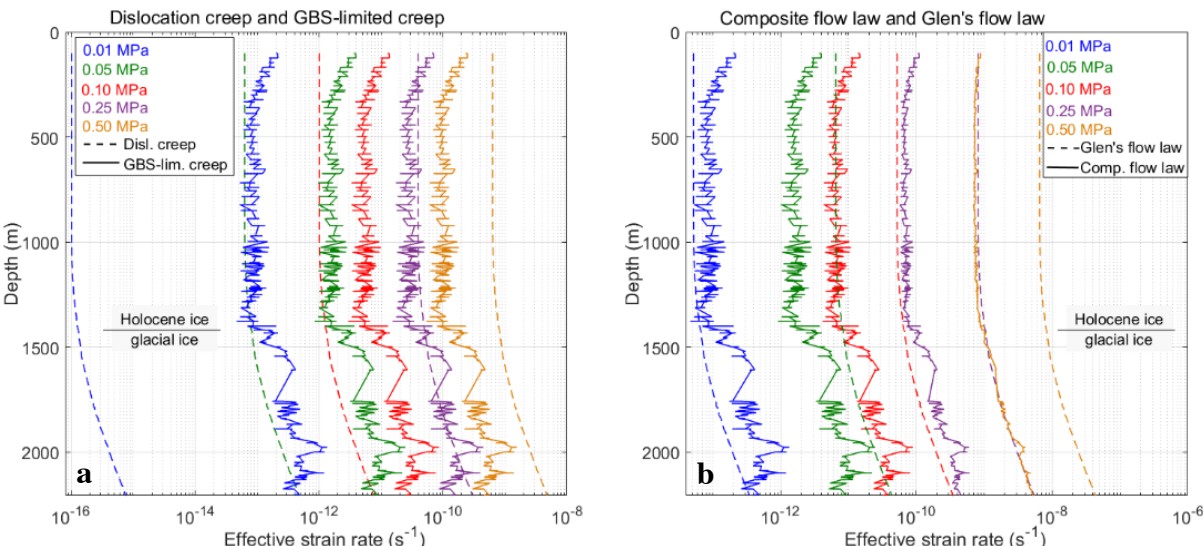

**Figure 10: Stress sensitivity of the deformation mechanisms in the Holocene and Glacial ice. (a) Strain rates predicted by the dislocation creep mechanism (dotted lines) and GBS-limited creep (continuous lines)** **at different constant stress values****. For readability, only the results of the micro-scale constant stress model are shown. (b) Plot of Glen's flow law (dotted lines) and the composite flow law (continuous lines). For (a) and (b) the temperature profile and the grain size distribution of the NEEM ice core were used in combination with the flow law parameters in Table 2.**