# Peer review of "Using a composite flow law to model deformation in the NEEM deep ice core, Greenland: Part 1 the role of grain size and grain size distribution on the deformation of Holocene and glacial ice"

_The Cryosphere, 2018_

## Referee Comment (RC1) · Anonymous Referee #1 · 19 Feb 2019

It is well-established that the flow of ice depends strongly on the crystallographic orientation. This paper completely ignores this and attempts to model ice sheet flow based on a dubious composite flow law, which considers only the grain size. I see little value in curve fitting with adjustable parameters to explain ice sheet flow.

---

## Author Comment (AC1) · 20 Feb 2019

The focus of this manuscript is to quantify the effects of grain size on flow of ice using a widely cited flow law with input from natural ice core data. The current study takes the additional step to examine the grain size distribution as a material state variable. On the basis of our models, we investigate and discuss various deformation mechanisms and estimate their relevance for bulk deformation, which is valid scientific practice.

Our models suggest small grains can explain the often observed increase in strain rate in finer grained layers. This result is very important and we consider it worthy of publication.

Work on the relevance of micro-scale processes on ice deformation is essential to improve our understanding of ice sheet flow, which is now possible with the wealth of data we have gathered with our new evaluation methods. R#1 states flow of ice is strongly dependent on anisotropy and yet anisotropy is unable to explain certain phenomenological observations on flow variations, such as the lack of correlation between borehole logging results and anisotropy variations (pers. comm. Dahl-Jensen 31.1.2019; Weikusat et al. 2017, Cuffey et al. 2000). These and other aspects of the effect of anisotropy are discussed in Section 5.5. We thank R1 that this point is not clear enough and will add additional citations / wording to support our arguments,

There is still much we do not understand about the rate controlling mechanisms of flow in ice. Using models and grain size data from naturally deformed ice is a valid approach to examine these mechanisms.

---

## Referee Comment (RC2) · David Prior (Referee) · 17 Apr 2019

Review of: **Kuiper**, Weikusat, de Bresser, Jansen, Pennock and Drury "Using a composite flow law to model deformation in the NEEM deep ice core, Greenland: Part 1 the role of grain size and grain size distribution on the deformation of Holocene and glacial ice"

By **Dave Prior** University of Otago.

This is an important paper and an excellent piece of science. The manuscript needs some significant modification to help readers understand the paper, to highlight its importance and improve potential impact. The significance of this paper is that it demonstrates that a composite flow law that involves both grain size sensitive and grain size insensitive deformation mechanisms can be used to model the behavior of polar ice deforming at natural conditions. Moreover the analysis suggests that under low stress conditions (ice divides) the grain size sensitive mechanisms could dominate the rheology.  I have also reviewed the part two paper and I think the authors decision to separate the two papers is a good one. The outcomes are clearer and impacts are more effective as two papers.

I also have an annotated pdf the authors can have.

**The mistake in the Goldsby & Kohlstedt composite flow law.**
The paper corrects a mistake in the dislocation creep component of the Goldsby & Kohlstedt composite flow law. The validity of the correction is nicely illustrated in Fig 6a. The way this correction is used is convoluted and I think a reader who is not aware of this issue will be thoroughly confused.  There are two problems:

1. The manuscript does not make it clear this is a mistake in the Goldsby & Kohlstedt data analysis, so the reader will be wondering is this just an alternative analysis. Maybe you are trying to be too polite: don't worry everyone makes errors and when we identify them they need to be corrected. You need to be absolutely upfront about this being a mistake. E.g. "We and other researchers (Goldsby pers comm, Prior pers comm) have spotted an error in the fit of the <258K dislocation creep flow law to the data in Goldsby and Kohlstedt. We have recalculated the flow law based on the original data...".
2. Figure 7 and the discussion around it are pointless. You are using a flow law (the original G&L <258K dislocation creep flow law) which you show is wrong. This serves no purpose and it just makes the paper really really confusing. You can make the general point that the re-fitted composite flow law tends to decrease the importance of dislocation creep relative to GBS, in the section where you discuss the re-fitting of the flow law.

I think you need short names that clearly distinguish the different flow law fits. This becomes particularly important when one considers the two parts of your work as the second paper has a different fit (for justifiable reasons). Something like

- G&K: original Goldsby and Kohlstedt flow laws.
- G&K$_{corr}$: Goldsby and Kohlstedt flow laws corrected as in part 1.
- G&K$_{262}$: Goldsby and Kohlstedt flow laws with best fit for 262K switch (related to the part 2 paper)

I'm sure you can do better than this suggestion- but it needs something otherwise we will all be very confused.

**A schematic overview at start**

The paper needs an introductory schematic overview figure of the microstructures and grain sizes in the NEEM core: basically, an annotated depth profile. Readers are busy and you cannot rely on them looking up the source literature so having this figure up front will increase impact and uptake.  Most readers will be unfamiliar with NEEM. The figure could include the T and grain size profiles currently in fig 7,8 (enabling these figs to be simplified) as well as the stratigraphic info.  The images in current fig 1 could potentially be incorporated in this.

**"Accommodated" by**

Expressions such as "grain boundary sliding accommodated by easy slip" are commonly used by the rock deformation community. The problem is that this terminology is not used consistently. I find this language highly uninformative. If it is used to indicate a mechanism dependency then which is the "dependent" mechanisms depends on how you understand the English: different readers interpret it in opposite ways. Furthermore some use this terminology to indicate the mechanism within the grain boundary (as opposed to a kinematically required partner mechanism) as discussed in some of the original GBS literature from Michael Ashby (see for example fig 6.1 in Schulson and Duval, 2009).  In your paper the language becomes particularly confusing through variation of language used - especially bearing in mind that many of the readers are not from the rock deformation community. This language discussion arises repeatedly and I recall a meeting back in 2006 where I was involved in extensive discussions with at least for the two co-authors on this topic.  There are several statements that inform what language might be useful:

- Grain boundary sliding of a polycrystalline material (where pore spaces are not allowed) requires that the individual crystals change shape.
- Where a polycrystalline deforms by a mechanism that restrict the shape change of each individual crystal (e.g. glide on one crystal plane and homogenous bulk strain), grain boundary sliding is required.
- Diffusion creep in a polycrystalline material requires grain boundary sliding.

You are primarily trying to explain the flow law form:

$$(\frac{1}{\dot{\varepsilon}_{basal}} + \frac{1}{\dot{\varepsilon}_{gbs}})^{-1}$$

embedded within equation (2).  The explanation on lines 6 and 7 of page 5 are not going to help the reader understand this. The way I usually explain this mechanism is that GBS is accompanied by basal slip. The two mechanisms are dependent upon each other, one cannot proceed without the other. The explanation on line 7 is particularly confusing as it indicates (wrongly) that both of the inverse terms inside the brackets each involves both basal slip and GBS.

$\frac{1}{\dot{\varepsilon}_{basal}}$ is just the inverse of the strain rate related to basal slip. GBS is not involved.

$\frac{1}{\dot{\varepsilon}_{gbs}}$ is just the inverse of the strain rate related to GBS. Basal slip is not involved.

It is the expression as a whole that provides the rate dependence. So that if

- $\dot{\varepsilon}_{basal} \gg \dot{\varepsilon}_{gbs}$ then $(\frac{1}{\dot{\varepsilon}_{basal}} + \frac{1}{\dot{\varepsilon}_{gbs}})^{-1} \approx \dot{\varepsilon}_{gbs}$ ie GBS limits the strain rate

- $\dot{\varepsilon}_{basal} \ll \dot{\varepsilon}_{gbs}$ then $(\frac{1}{\dot{\varepsilon}_{basal}} + \frac{1}{\dot{\varepsilon}_{gbs}})^{-1} \approx \dot{\varepsilon}_{basal}$ ie basal slip limits the strain rate.

You use the "rate limiting" terminology (in addition to the accommodation terminology) and this language is much more satisfactory to me. I think that you can make the paper much clearer by abandoning the "accommodated by" expression and describing the mechanism balance in terms of rate limits. In the discussion around lines15 to 21 on page 5 you could usefully incorporate the two bullet points listed above. That then gives a much clearer basis for the simplification to equation 4.

**The "Glen" law**
I think you need to take care with the language used related to the Glen law. Citing Glen (1955) for a Glen law with n=3 does a disservice to John Glen. Glen's three key papers have n values of 4 (1952), 3.3 changing to 4 (1953) and 3.2 to 4.2 (1955) respectively. As far as I know Glen has not written that one should use an n=3 relationship; if anything, he suggests that n values for naturally deforming ice should be around 4. So, the n=3 is a simplification of Glen's work that is in common use (I'm not really sure who did this first). The Glen law in common use has n=3 but it was not Glen who set this value. It would be nice if your introduction of the Glen law made this subtlety clear.

**Discussion**
The discussion is too long and rather rambling. I have some specific suggestions that follow but I would suggest some significant shortening beyond these points. A rambling discussion just weakens a paper's impact.

**Put all the discussion of the modified flow law in one place**

As commented earlier, the modification is to correct an error. So it is not really a discussion point. Put all or the discussion of this issue in the text where the error is corrected. E.g. move page 11 L8-L14 to earlier.

**Put the discussion of the micro scale constant stress and constant strain rate models in one place.**

This is an excellent piece of work, but looses coherence by being spread through the manuscript. I would suggest that fig 9 and ensuing discussion goes before figure 8 (figure 7 should be axed). This will make the paper easier to follow and will mean an explanation is already at hand for the strain rate variation of dislocation creep in fig 8.

**Grain size: mean diameter vs mean area**

The exploration of using grain size distributions rather than means in flow laws is excellent. One thing that is probably worth mentioning is that the convention in the glacial literature is to use the mean area. Since this is measured by counting the number of grains in an area, backing out a mean diameter is more or less impossible (needs standard deviation of the normal distribution to do this). For normal distributions of diameter or of log diameter (as is common for recrystallized grain size distributions from experiments) the equivalent diameter calculated from mean area will be larger than the mean diameter. Application of the GSS flow law elements to mean area data would need this to be considered.

**Girdle**

When you use the term girdle to describe a CPO element can you describe this more completely. Girdle covers a wide range of things on a stereonet. I restrict the term for great circle distributions, but many include small circle distributions under this name. Even if more restricted some information on "girdle" orientation would be useful.

**Recovery and recrystallisation**

These are very important processes in deforming glacial ice. They get very little space in this paper?

**Strain rate**

The layer thinning basis (page 7 line 32-33) for strain rate estimates needs explaining a bit more completely so the reader understands the basis of the strain rate estimates.

**CPOs during GBS in ice.**

The discussion on page 14 lines 23-25ish could make reference to a paper by one of my students.(Craw et al., 2018) show incredibly strong CPOs develop at relatively low strain (20% shortening) in large grains. In this case the large

grains are not strongly strained (they do not have elongated shapes) and the large grains are surrounded by a network of fine recrystallized grains that have an equivalent but much weaker CPO. In that paper we suggest that GBS is an important mechanism controlling the microstructural evolution but some slip on the basal plane of the large grains is needed to develop such a strong CPO.

**Figure Captions**
Generally figure captions are way too long and include discussion elements that should be in the main text. The role of the figure caption should be to explain what is in the figure, where that is not clear from the figure itself. Discussion of the significance of a figure should be in the text.

**Figure 8 layout**
The layout of figure 8 can be improved significantly.
- If the G-size and temperature are in a schematic at the start of the paper they can be omitted here. The GBS and composite flow laws mirror the G-size profile so well that it does not need to be on the same fig. Similarly the acceleration at the bottom of the hole clearly corresponds to temperature so the depths and the stratigraphic labels give enough cross reference.
- The reason for removing T and G-size is that a much neater figure is possible if you stack a,b and c vertically above each other. This makes the strain rate position of lines much easier to compare.
- Label the axis of the right-hand graph as "GBS contribution (%)" rather than the label "percentage".
- Make all the lines solid (dashed lines do not work for wiggly lines) and label them with rotated vertical text next to the line, in the same colour as the line. This and the last point mean that you can get rid of the boxed legend.
- Make the colours bold and clear. The yellow (GBS) is not good.

**Some refs I think you should have in there:**
(Durham and Goetze, 1977; Durham et al., 2010; Durham et al., 2001; Pettit and Waddington, 2003; Pettit et al., 2011)

Craw, L., Qi, C., Prior, D. J., Goldsby, D. L., and Kim, D., 2018, Mechanics and microstructure of deformed natural anisotropic ice: Journal of Structural Geology, v. 115, p. 152-166.

Durham, W. B., and Goetze, C., 1977, Plastic-flow of oriented single-crystals of olivine .1. mechanical data: Journal of Geophysical Research, v. 82, no. 36, p. 5737-5753.

Durham, W. B., Prieto-Ballesteros, O., Goldsby, D. L., and Kargel, J. S., 2010, Rheological and Thermal Properties of Icy Materials: Space Science Reviews, v. 153, no. 1-4, p. 273-298.

Durham, W. B., Stern, L. A., and Kirby, S. H., 2001, Rheology of ice I at low stress and elevated confining pressure: Journal Of Geophysical Research-Solid Earth, v. 106, no. B6, p. 11031-11042.

Pettit, E. C., and Waddington, E. D., 2003, Ice flow at low deviatoric stress: Journal of Glaciology, v. 49, no. 166, p. 359-369.

Pettit, E. C., Waddington, E. D., Harrison, W. D., Thorsteinsson, T., Elsberg, D., Morack, J., and Zumberge, M. A., 2011, The crossover stress, anisotropy and the ice flow law at Siple Dome, West Antarctica: Journal of Glaciology, v. 57, no. 201, p. 39-52.

---

## Author Response (AR1)

Response to comments of referee #2 (David Prior)

We would like to thank the referee for their detailed comments on the paper, which are perceptive and very helpful. We have largely implemented the suggestions from the referee in the revised manuscript.

**Referee's first comment**
*The mistake in the Goldsby & Kohlstedt composite flow law*
**Authors response** We acknowledge that the way this correction was written down could be confusing to readers of the paper. We added in section 3 that the discrepancy between the results of the calculated strain rate and the experimental data points was identified and confirmed by Goldsby and Prior.
Figure 7 was removed from the manuscript and all discussion about the flow law parameters for dislocation creep was moved to section 3. In the end of section 3 it was also added that dislocation creep is 15 to 20 times slower with the new flow law parameters compared to the old flow law parameters.
The flow law parameters for dislocation creep in Table 2 have been named "G&K$_{mod}$ dislocation creep" in order to distinguish between the correction made to the flow law parameters in this paper and the companion paper (tc-2018-275).

**Referee's second comment**
*A schematic overview at start*
**Authors response** A schematic overview was added in the beginning of the paper. The figure shows (a) the three CPO eigenvalues, (b) the grain area and (c) the in-situ temperature. The figure also shows the three depth regimes (Holocene, glacial and Eemian ice) in the NEEM ice core. The grain size and temperature data was taken out of Figure 7 and 8 (of which only figure 8 is left now).

**Referee's third comment**
*"Accommodated" by*
**Authors response** The explanation on line 7 has been removed and this has been corrected for. Throughout the entire paper (and the companion paper tc-2018-275) we have adopted the 'rate limiting' terminology instead of the 'accommodated by' terminology. The two bullet points are incorporated in the methods now (Equation 4 and 5 in the new version).

**Referee's fourth comment**
*The "Glen" law*
**Authors response** We mentioned that 'the most often used for of Glen's flow law' has a value of n=3. This is introduced just below Equation 1. The value of n=3 was taken from Paterson (1994) and has been cited accordingly.

**Referee's fifth comment**
*Discussion*
**Authors response** The discussion has been shortened. Most of the shortening was accomplished by moving the discussion about the flow law constants for dislocation creep to section 3 and removing the discussion of the results obtained using the original flow law constants (Figure 7 in older version).

**Referee's sixth comment**
*Put all the discussion of the modified flow law in one place*

**Authors response** All the discussion of the modification of the flow law parameters for dislocation creep has been moved to section 3.

**Referee's seventh comment**
*Put the discussion of the micro scale constant stress and constant strain rate models in one place.*
**Authors response** Figure 9 (Figure 8 in new version) has been put before Figure 8 (Figure 9 in new version). The results and discussion have been reordered accordingly.

**Referee's eighth comment**
*Grain size: mean diameter vs mean area*
**Authors response** We have mentioned that the equivalent diameter calculated from mean area is larger than the mean diameter in section 2.3 on page 7 lines 23-26.

**Referee's ninth comment**
*Girdle*
**Authors response** We described the shape of the CPO in the Holocene ice as 'great circle "girdle" distribution' (page 4 line 19). The CPO eigenvalues (Figure 1a in new version) will also help to clarify the type of the CPO.

**Referee's tenth comment**
*Recovery and recrystallization*
**Authors response** *we agree that Recovery and recrystallization are important processes in glacial ice. In section 5.6 we suggest that dynamic recrystallization by SIBM is one reason why the predicted deformation mechanisms from the composite flow law models are not consistent with the microstructures developed in the Holocene ice. The role of SIGM is discussed extensively in the manuscript and we have now added the wording of "recrystallization by SIBM".*

**Referee's eleventh comment**
*Strain rate*
**Authors response** *The methods are described in the references of Gillet-Chaulet et al., 2011; Montagnat et al., 2014, so we have not added any further details here.*

**Referee's twelfth comment**
*CPOs during GBS in ice*
**Authors response** The reference to Craw et al. (2018) was added to the manuscript in section 5.5.

**Referee's thirteenth comment**
*Figure Captions*
**Authors response** On a few occasions a sentence was removed in the figure caption. However, for most of the figures the caption did not change. For these figures we think that the caption explained well what was in the figure and shortening the caption would have been undesirable.

**Referee's fourteenth comment**
*Figure 8 layout*
**Authors response** The comments in all five bullet points were included when adjusting the figure. We agree that these adjustments make the figure clearer and easier to interpret.

**Referee's fifteenth comment**

*Some refs I think you should have in there:*

**Authors response** Four out of the six papers were cited were appropriate and added to the reference list in the manuscript.

---

## Referee Report (RR1)

This paper is based around the hypothesis involving grain boundary sliding (GBS) presented by Goldsby & Kohlstedt (1997; 2001). Goldsby & Kohlstedt initially didn't describe a flow law for ice in their early papers, instead they made a proposition (e.g. 1997, p.1404) "we suggest that GBS accounts for the remaining strain" this is supported by one micrograph which shows grain sizes of ~20 microns (Fig. 6 in Goldsby & Kohlstedt 1997). This is considerably finer than any of the <1 mm NEEM ice illustrated in Figures 2 and 3 in the present paper and the micrographs in Fig. 1 in the Part 2 paper. Using the Goldsby & Kohlstedt assumptions and hypothesis the current authors are presenting two different flow models, based on arguments that came from originally low strain experiments (~0.2 strain). Whereas in nature strains are substantially larger and represent ice deforming well into a Tertiary creep regime, something that is difficult to replicate in a laboratory. There are a large number of assumptions being made by the current authors concerning deformation mechanisms and processes producing the grain sizes that we see in the NEEM ice core (page 7) and the shear stress (page 8). Nowhere in this paper were modelled strain rates (Figs 9 and 10) compared with real closure rates data (strain rates) obtained during or after drilling. It is well known that in many ice sheets there are large strains and significant strain rate variations with depth and laterally as pointed out by Bons et al. (2018) and very many other authors; this has also been recognized in other drill sites on ice divides. Therefore it is hard to see how these strain rates are going to be replicated particularly if the authors assume GBS is critical to their models. The authors assume a constant stress of 0.07 MPa along the length of the NEEM ice cores (page 8 line 28) which is highly unlikely. The aims of the paper are summarised in a statement on page 9, namely:

"As this paper explores the effect of grain size, grain size distribution and different micro-scale models on the dominant deformation mechanism and the total strain rate, it is beyond the scope of this study to derive a stress-depth model for NEEM because this requires knowledge on the rheology, which is the property that is investigated here."

Well this is a very unclear statement and aim should have been clearly stated earlier in the paper. What is meant by "total strain rate"? Surely there must also be a discussion of the rheology? If rheology is investigated here, then why is it not described in this paper? Later on the authors foreshadow a companion paper (Part 2) on rheology (page 9, line 25; but this only considers the bottom section of the ice core); you can't separate this out from the present study. It would be sensible to combine these two papers into one seminal paper.

**Title of paper is misleading and too long** – There should be no Part 1 but combine it with a Part 2, this will avoid a lot of repetition, deficiencies in Part 1 such as the lack of proper discussion on role of temperature, discussion of CPOs and clearly bring out the differences above and below the 2000 metre depth in the NEEM ice cores. Words such as "on the deformation of Holocene and glacial ice" should be deleted from title as the general reader won't understand that modelling is only undertaken in this one section and not along the total ice core length.

**Abstract:** The last sentence is misleading as this paper in its current form is not discussing rheology. Again I feel that by combining Parts 1 and 2 into one paper would avoid some repetition.

**The introduction** nicely summarises ideas concerning GSI vs. GSS mechanisms and discusses grain size distributions. However, the authors really ignore the grain scale processes that are described by many other authors such as the competition between new grain nucleation vs. grain boundary migration. They even state that they are ignoring this in their response to Referee 1. This is a very major failing of the current paper. Although there is some discussion relevant to this on page 17.

The last section in the 'introduction' is about grain sizes, which the authors consider to be the primary controlling entity on strain rates in natural ice. Instead these are a product of different strain

rates, strain histories and temperature and it is tantamount to a conspiracy that the author's dismiss (page 4 line 5) observations from previous NEEM workers to immediately assume that GSS mechanisms may dominate the deformation through a 2000+ metre ice column.

In fact I find this 'Introduction" is far too long and could be appreciably shortened, it is verbose and this is a problem throughout the paper.

**Discussion on grain sizes** (e.g. page 3, 5, 6 and 7) and late reference to Fig. 5 (Page 7) could be better consolidated rather than being variously discussed in different parts of the text, with some of the text incorporated into expanded captions for figures 4 and 5.

**Methods:** The application of GBS processes in ice and the predictions and hypothesis of Goldsby & Kohlstedt (2001) have for many researchers always been considered speculative and have been seriously questioned by numerous researchers in the glaciology community (starting with Duval & Montagat, 2002). The existence of grain boundary sliding at high strains as described in the concluding paragraph of the 1997 paper of Goldsby & Kohlstedt have not been convincingly identified in other experiments nor in nature. During the revisions to this paper the authors inserted an inappropriate reference to Craw et al. (2018) on page 15 to support GBS; in the latter case it involved strain localisation and may be a case as described by Duval and Lliboutry (1985). Even if we go back to Duval & Lliboutry's 1985 paper, they suggest grain growth is important relieve to any GBS and the latter is not a dominant mechanism.

Referee 1 questions the lack of taking into the account of CPO development in their models. The authors dismiss this (page 10, lines 1-5) and in section 5.5. In section 5.5 the author's argue that GSS is important. However the authors fail to present any glaciological or experimental study using coarser ice that clearly illustrates this is an important and known phenomena particularly in the high strains and temperatures observed in ice sheets. The majority of previous assertions for GBS are based on theoretical grounds, the questionable interpretation of Goldsby & Kohlstedt and do not appear to be a reality in natural ice (e.g. Duval & Montagnat 2002).

This composite flow law model (page 5) for ice needs to be better supported by definitive ice experiments, but a lot of the cited references are based on experiments that researchers had no idea of the true grain scale processes.  In the results section, Fig 8 should be compared to those of the large strain experiments of Peternell et al. (2019) where a steady state rheology was only reached in fast strain rate experiments. In contrast, during a slow deformation there is insufficient seeding of new grains to enable continuous recovery, and there is a bimodal grain size distribution, but no evidence of GBS. These results are explained by stress concentrations within grains and not GBS. Particularly where basal planes are in unfavourable orientations for basal slip (hard glide orientation) coinciding with the development of a bimodal grain size distribution. In this and may other in situ experiments GBS is not observed in a polycrystalline aggregate, except under very special circumstances where grains may be highly anisotropic.

**Page 4 line 6:** I take exception to the statement "while the microstructures in the glacier ice suggest that GSS mechanisms may be important" This is not true and is not accepted by most members of the ice community.

**2.1 Study site:** There is overlap between Part 1 and the Part 2companion paper. It would be good if these could be combined and much of the unnecessary detail in Part 2 be removed. I also feel that **section 2.3** boundary conditions could be included in this section. A more concise description could be achieved by amalgamating these two parts.

**2.2 Composite flow law**: Given my scepticism about the composite flow law this section is well argued. However, I would like to see more discussion about other flow laws e.g. Glens as this is what everything is being compared against and is better expressed in Part 2.

**Section 4 _ Results**: Initially I found this section difficult to read and verbose, but after reading **section 5.1** some clarification occurred (Except delete discussion about Olivine). It may be worthwhile to reword and shorten this section.

**Section 5.3**: In reading this section and comparing figures 10a with 10b the significant differences, implications and the derivation of the stress exponent between these models are not clearly enunciated in the first part of this discussion.  I believe with rearranging this section it will become clearer to the reader. Also could these two figures be combined as there is overlap?

**Section 5.5.**

This section in its present form should be deleted. Much of the discussion here is about other mineral systems and is irrelevant to the present models. There is substantial overlap with what is said in the following section. Only statements such as below should go into a reworded introduction.

"The c-axis eigenvalues show a minor variability in the glacial ice of the NEEM ice core (Eichler et al., 2013; Montagnat et al., 2014) where the layers of high strain rate are predicted. The strong development of CPO and the development of 10 substructures with depth indicate that large amounts of strain are accommodated by basal slip of dislocations in the NEEM ice core."

Alternatively, if Parts 1 and 2 are combined into a single paper then the authors would address the concerns of Referee 1. The statement at the top of page 10 is vague

**Section 5.6:** The overall statements in this section are almost contradictory with the last paragraph of this section really invalidating some of the reasons of applying these GBS-limited models.

**The conclusions** to this paper rely solely on strain models and the effect of a temperature input is only mentioned briefly in results (page 10. Line 30); this is really left to Part 2. There is no discussion of possible limitations in the application of these models, except in **section 5.6** and this should have been better integrated with the conclusions. There should be a discussion and comparison to what is happening in the lower levels of the ice core but this is left to Part 2.

I would strongly recommend papers 1 and 2 are combined. A shorter more concise paper will have a far bigger impact. For instance a good comparison between differences between Fig. 10 in Part 1 Vs. Fig. 4 in Part 2 would better explain the differences different models as they relate to temperature and strain regimes. It would have been good if the scales for Effective strain rates were the same.

Chris Wilson

16/10/2019

**Additional references**

Duval , P. Lliboutry L. 1985. Superplasticity owing to grain growth in polar ices. J. Glaciol. 31 60-62.

Duval, P. & Montagat, M. 2002. Comment on Goldsby & Kohlsted -  J Geophy Res. V107, B4 2082

Peternell, M., Wilson, C.J.L., and Hammes, D.M., 2019. Strain rate dependence for evolution of steady state grain sizes: Insights from high-strain experiments on ice. *Earth and Planet. Sci. Lett.,* 506, 168-174.

---

## Referee Report (RR2)

Review after Author revisions of: **Kuiper**, Weikusat, de Bresser, Jansen, Pennock and Drury "Using a composite flow law to model deformation in the NEEM deep ice core, Greenland: Part 1 the role of grain size and grain size distribution on the deformation of Holocene and glacial ice"

The paper is much improved. A supurb contribution. It is now easier to follow and I think will have a greater impact. A have a few suggestions and comments to improve things further.

**The Mistake in G&K**
The opening two paragraphs of section 3 (flow law parameters) are much better- but they still skirt around the issue that this is an unfortunate mistake and a reader has to work really hard to work this out. I think this can be corrected by some minor changes:

Line 10: Add a new sentence after "..experimental data": "This analysis highlighted an error in one part of the published flow law. The error was confirmed by others (Goldsby pers comm; Prior Pers comm) and is explained and corrected in the next two paragraphs". Then start a new paragraph with "A comparison…."

Line 21,22: Remove most of these two lines and start the paragraph with "We modified the flow law…."

**Minor things**

Page 1. Line 15: comma after slip. Hyphen between rate and limited (and if you do hyphenate- do for whole manuscript).

Page 1. Line 16: comma after slip. Hyphen between rate and limited.

Page 2. Line 5: Remove GBM from the list- it is not a deformation mechanism. After the citations on line 7 you could add "… and associated with recrystallisation mechanisms, including GBM."

Page 3. Line 11: Replace "only one defm mech, that is" with "just".

Page 3. Line 13: Key reference for n=4 at Tertiary creep would be (Durham et al., 1983).

Page 3. Line 22: The new bit in red does not make that much sense. I think the alternative you are trying to suggest is a flow law that reflects directly the behavior at steady state or in Tertiary creep. Evidence that this has a different stress exponent in (Glen, 1953, 1955; Qi et al., 2017; Treverrow et al., 2012).

Page 9. Line 30: Remove "Therefore".

Page 14. Line 3: The low stress different n value is also shown and commented on in (Bons et al., 2018).

Page 16. Line 7: I don't think you have spelt out "SIBM". Earlier in the manuscript (see page 2 comment) you use GBM. Maybe replace GBM with SIBM or better with SIGBM. Just pick one to be consistent and then change all occurrences.

Page 17. Line 20:   Remove the first sentence. Results using a flow law shown to be wrong are irrelevant.

Page 17. Line 31.   I think the term "margins" will be confusing here. You have used margins earlier and so the context is set. Is it also worth mentioning other potentially high stress settings such as basal zones and lateral shear margins?

Bons, P. D., Kleiner, T., Llorens, M. G., Prior, D. J., Sachau, T., Weikusat, I., and Jansen, D., 2018, Greenland Ice Sheet: Higher Nonlinearity of Ice Flow Significantly Reduces Estimated Basal Motion: Geophysical Research Letters, v. 45, no. 13, p. 6542-6548.

Durham, W. B., Heard, H. C., and Kirby, S. H., 1983, Experimental deformation of polycrystalline h2o ice at high-pressure and low-temperature - preliminary-results: Journal of Geophysical Research, v. 88, p. B377-B392.

Glen, J. W., 1953, Rate of flow of polycrystalline ice: Nature, v. 172, no. 4381, p. 721-722.

-, 1955, The creep of polycrystalline ice: Proceedings of the Royal Society of London Series a-Mathematical and Physical Sciences, v. 228, no. 1175, p. 519-538.

Qi, C., Goldsby, D. L., and Prior, D. J., 2017, The down-stress transition from cluster to cone fabrics in experimentally deformed ice: Earth and Planetary Science Letters, v. 471, p. 136-147.

Treverrow, A., Budd, W. F., Jacka, T. H., and Warner, R. C., 2012, The tertiary creep of polycrystalline ice: experimental evidence for stress-dependent levels of strain-rate enhancement: Journal of Glaciology, v. 58, no. 208, p. 301-314.

---

## Author Response (AR2)

Response to Referee Report 1 by Chris Wilson.

We thank the referee for the detailed and helpful comments in the review. We have revised the paper substantially to take account of the comments. The referee comments are listed below with our response in blue text after the comment. All references mentioned in our response are in the reference list of our manuscript.

This paper is based around the hypothesis involving grain boundary sliding (GBS) presented by Goldsby & Kohlstedt (1997; 2001). Goldsby & Kohlstedt initially didn't describe a flow law for ice in their early papers, instead they made a proposition (e.g. 1997, p.1404) "we suggest that GBS accounts for the remaining strain" this is supported by one micrograph which shows grain sizes of ~20 microns (Fig. 6 in Goldsby & Kohlstedt 1997). This is considerably finer than any of the <1 mm NEEM ice illustrated in Figures 2 and 3 in the present paper and the micrographs in Fig. 1 in the Part 2 paper.

We agree that Goldsby and Kohlstedt worked on a much finer grain size than occur in polar ice (Figure 2): however, GSS creep can be dominant in materials with larger grain size when stress levels are low, as is the case for the NEEM core data. The experiments of Goldsby & Kohlstedt were conducted at higher stresses than in natural ice sheets, so the transition to GSS creep is expected to occur at larger grain sizes in natural materials. This point is clearly illustrated in the deformation mechanism maps for ice (Durham et al., 2010) as shown below. According to this map GSS creep is dominant at grain sizes less than about 1mm, when stress is less than 1 MPa.

[Figure]

Figure 6 from Durham et al. (2010)

Furthermore, the model we applied takes into consideration not only the mean grain size, but the full grain size distribution, which can have a large effect on the occurrence of GSS or GSI creep mechanisms (Ter Heege et al., 2004).

Using the Goldsby & Kohlstedt assumptions and hypothesis the current authors are presenting two different flow models, based on arguments that came from originally low strain experiments (~0.2 strain). Whereas in nature strains are substantially larger and represent ice deforming well into a Tertiary creep regime, something that is difficult to replicate in a laboratory. There are a large number of assumptions being made by the current authors concerning deformation mechanisms and processes producing the grain sizes that we see in the NEEM ice core (page 7) and the shear stress (page 8).

The referee makes an important point concerning the uncertainties about the deformation mechanisms involved in fine-grained ice and about the possible role of tertiary creep.

Concerning creep of fine-grained ice, there is very little data on the experimental deformation of fine-grained ice. A welcome recent addition is the study by Saruya et al. (2019) who have confirmed grain size sensitive creep. This reference has been added (e.g. page 3) to highlight this point. As yet, there are no large strain deformation studies in this regime. We have added the following sentences in the methods to account for the referee's comment. "the mechanisms involved in the low stress, grain size sensitive deformation regime in ice are debated and several different creep models have been proposed (Montagnat and Duval 2000; Goldsby and Kohlstedt 2001; Saruya et al. 2019). While the deformation mechanisms involved are controversial there is some consensus from the experimental studies on the flow law parameters of this regime (Goldsby and Kohlstedt 2001; Saruya et al; 2019) with n about 2 and p about 1.4. As we are using the simplified version of the composite flow law in equation (6) our study is not dependent on the details of the deformation mechanisms in the GSS regime".

Concerning the possible role of tertiary creep, as noted by the referee, strains in polar ice are substantially higher than in experiments, so natural ice may be expected to deform in the tertiary creep regime. The flow laws of Goldsby and Kohlstedt (1997, 2001) are for secondary creep which may not be relevant at high strains. However, this is also the case for Glen's flow law so this is a potential limitation for all ice flow models. We note that the recent experiments of Saruya et al. (2019) show no evidence of tertiary creep during GSS deformation to modest strains.

Nowhere in this paper were modelled strain rates (Figs 9 and 10) compared with real closure rates data (strain rates) obtained during or after drilling. It is well known that in many ice sheets there are large strains and significant strain rate variations with depth and laterally as pointed out by Bons et al. (2018) and very many other authors; this has also been recognized in other drill sites on ice divides. Therefore, it is hard to see how these strain rates are going to be replicated particularly if the authors assume GBS is critical to their models. The authors assume a constant stress of 0.07 MPa along the length of the NEEM ice cores (page 8 line 28) which is highly unlikely.

The variation of strain rate with depth is currently being investigated by long term bore-hole logging studies (Dahl-Jansen et al. 2016) at NEEM. This study is in progress and will provide important information on the effect of grain size, CPO and impurity content on ice deformation. Preliminary results from the bore-hole logging (Greve et al. 2017; Dahl Jansen, pers. comm 2019). Indicate enhanced deformation in the Glacial age ice at NEEM. The prediction of our paper is that fast strain rates will be found in the finest-grained layers of the NEEM glacial ice (1409-2207). In this depth range the ice has uniformly strong CPOs, so zones of fast strain rate within this depth range may be explained by grain size or impurity effects but cannot be explained by variations of anisotropy. We have added some additional references to section 5.4 which mention the potential reasons for enhanced strain rate in the Glacial ice.

The aims of the paper are summarised in a statement on page 9, namely:
"As this paper explores the effect of grain size, grain size distribution and different micro-scale models on the dominant deformation mechanism and the total strain rate, it is beyond the scope of this study to derive a stress-depth model for NEEM because this requires knowledge on the rheology, which is the property that is investigated here."
Well this is a very unclear statement and aim should have been clearly stated earlier in the paper. What is meant by "total strain rate"? Surely there must also be a discussion of the rheology? If rheology is investigated here, then why is it not described in this paper? Later on the authors foreshadow a companion paper (Part 2) on rheology (page 9, line 25; but this only considers the bottom section of the ice core); you can't separate this out from the present study. It would be sensible to combine these two papers into one seminal paper.

We have moved and rewritten the paragraph about the aim of the study. Concerning the presentation of the work in two parts, we have retained the structure of two companion papers, particularly because the second referee as well as the editor clearly support this subdivision. Combining the two papers would avoid some repetition and bring out the differences above and below the 2000m depth, but the present structure has the advantage of first defining modified composite flow laws for ice, without needing to consider the complications of pre-melting. Each part focuses on particular effects. So part I focuses on the effects of grain size and grain size distribution, demonstrating that grain size distributions do not have a major effect. In part 2 we consider the effects of grain size and pre-melting. Each of these effects, grain sizes and pre-melting, deserves proper consideration in a separate paper.

**Title of paper is misleading and too long** – There should be no Part 1 but combine it with a Part 2, this will avoid a lot of repetition, deficiencies in Part 1 such as the lack of proper discussion on role of temperature, discussion of CPOs and clearly bring out the differences above and below the 2000 metre depth in the NEEM ice cores. Words such as "on the deformation of Holocene and glacial ice" should be deleted from title as the general reader won't understand that modelling is only undertaken in this one section and not along the total ice core length.

We have changed the title as suggested to remove the reference to Holocene and glacial ice. We have retained the structure of two companion papers, particularly because the second referee as well as the editor clearly support this subdivision

**Abstract:** The last sentence is misleading as this paper in its current form is not discussing rheology. Again I feel that by combining Parts 1 and 2 into one paper would avoid some repetition.

Our paper is about using a composite flow law to predict the strain rates in the NEEM borehole as a function of stress, temperature and material properties, so we think the paper is about the rheology. To avoid any misunderstanding, the last sentence in the abstract has been modified by replacing rheology, by composite flow law model. As noted above we feel that presenting the research in two parts is important to give full consideration of the different effects that can influence grain size sensitive flow in ice.

**The introduction** nicely summarises ideas concerning GSI vs. GSS mechanisms and discusses grain size distributions. However, the authors really ignore the grain scale processes that are described by many other authors such as the competition between new grain nucleation vs. grain boundary migration. They even state

that they are ignoring this in their response to Referee 1. This is a very major failing of the current paper. Although there is some discussion relevant to this on page 17.

The last section in the 'introduction' is about grain sizes, which the authors consider to be the primary controlling entity on strain rates in natural ice.

5   We agree that the grain scale processes that control grain sizes and grain size distributions are very important and any complete model for flow of ice will need to include evolution laws for these parameters. However, from our point of view, as a first approach to tackle GSS flow in nature, we rather focus on the effect of the grain size variations, not so much on the causes of the grain size variations. Indeed, a proper description of grain size evolution in natural ice is extremely important to further apply composite flow laws. Such a

10   description for "dynamic grain growth" under warm conditions is currently missing, and is not straightforward, e.g. due to the competing effects mentioned by the referee. We added some key references on recrystallization and grain size evolution in ice to acknowledge this point. As we have used the current grain size data from the NEEM ice core the model will only be relevant to this location of the Greenland ice sheet. We feel this is a useful first step in considering if GSS creep processes are important polar ice flow.

Instead these are a product of different strain rates, strain histories and temperature and it is tantamount to a conspiracy that the author's dismiss (page 4 line 5) observations from previous NEEM workers to immediately assume that GSS mechanisms may dominate the deformation through a 2000+ metre ice column. In fact I find this 'Introduction" is far too long and could be appreciably shortened, it is verbose and this is a problem

20   throughout the paper.

We agree that grain sizes are not fixed and are a product of the thermo-mechanical history. We added some key references on recrystallization and grain size evolution in ice to acknowledge this point. The sentence on page 4 line 5 does not dismiss any previous work and we definitely did not assume that GSS dominates throughout the NEEM column. By Glacial age we mean, the ice that was deposited as snow in the last glacial

25   maximum, which occurs in NEEM at depths of 1419 to 2207. We have altered the wording in the abstract and throughout the paper to make this clearer. To avoid any misunderstanding, we have expanded and rewritten this paragraph to include more studies on other Greenland ice cores as well as the NEEM papers that we already mentioned.

30   **Discussion on grain sizes** (e.g. page 3, 5, 6 and 7) and late reference to Fig. 5 (Page 7) could be better consolidated rather than being variously discussed in different parts of the text, with some of the text incorporated into expanded captions for figures 4 and 5.

As the effect of grain size is the main topic being considered in our paper, a discussion on grain size is needed in each section.

**Methods:** The application of GBS processes in ice and the predictions and hypothesis of Goldsby & Kohlstedt (2001) have for many researchers always been considered speculative and have been seriously questioned by numerous researchers in the glaciology community (starting with Duval & Montagat, 2002). The existence of grain boundary sliding at high strains as described in the concluding paragraph of the 1997 paper of Goldsby &

40   Kohlstedt have not been convincingly identified in other experiments nor in nature.

To take account of this comment and discuss the criticism of the predictions and hypothesis of Goldsby & Kohlstedt (2001), we have added a paragraph in the introduction page 3 line 19-31 and a new paragraph in the methods page 7 line 15-20 While the detailed deformation mechanisms are controversial, there is a

consensus that the low stress deformation of ice, involves slip on the easy basal slip system, accommodated or enhanced in some way by grain boundary processes (Goldsby 2006, Duval and Montagnat 2002, Saruya et al. 2019).

5   During the revisions to this paper the authors inserted an inappropriate reference to Craw et al. (2018) on page 15 to support GBS; in the latter case it involved strain localisation and may be a case as described by Duval and Lliboutry (1985). Even if we go back to Duval & Lliboutry's 1985 paper, they suggest grain growth is important relieve to any GBS and the latter is not a dominant mechanism.

We have removed the reference to Craw et al. (2018). The paper reports the occurrence of a strong
10  CPO in ice interpreted to deform by significant GBS, although the strong CPO is carried by the large grains and the finer-recrystallized grains interpreted to be deforming by GBS have a weaker CPO.  Given that other interpretations may be possible we did not include the reference.

Referee 1 questions the lack of taking into the account of CPO development in their models. The authors
15  dismiss this (page 10, lines 1-5) and in section 5.5.

We do not dismiss the comment in RC-1. On page 11 line 16-20 we point out that all deformation mechanisms considered in our paper may be enhanced by the presence of a CPO. We are unable to include the effect of CPO on the GSS mechanisms, because there is currently no experimental data available on the effect.

20  In section 5.5 the author's argue that GSS is important. However the authors fail to present any glaciological or experimental study using coarser ice that clearly illustrates this is an important and known phenomena particularly in the high strains and temperatures observed in ice sheets. The majority of previous assertions for GBS are based on theoretical grounds, the questionable interpretation of Goldsby & Kohlstedt and do not appear to be a reality in natural ice (e.g. Duval & Montagnat 2002). This composite flow law model (page 5) for
25  ice needs to be better supported by definitive ice experiments, but a lot of the cited references are based on experiments that researchers had no idea of the true grain scale processes. In the results section, Fig 8 should be compared to those of the large strain experiments of Peternell et al. (2019) where a steady state rheology was only reached in fast strain rate experiments. In contrast, during a slow deformation there is insufficient seeding of new grains to enable continuous recovery, and there is a bimodal grain size distribution, but no
30  evidence of GBS. These results are explained by stress concentrations within grains and not GBS. Particularly where basal planes are in unfavourable orientations for basal slip (hard glide orientation) coinciding with the development of a bimodal grain size distribution. In this and may other in situ experiments GBS is not observed in a polycrystalline aggregate, except under very special circumstances where grains may be highly anisotropic.

We have rewritten section 5.5 substantially to account for these comments. We have expanded the
35  discussion of the microstructural evidence for deformation mechanisms in the NEEM core. We point out that the presence of a CPO does not necessarily rule out grain size sensitive flow and refer to new experiments by Saruya et al. (2019) who propose a different mechanism for GSS creep in ice, involving dislocation creep enhanced by the presence of grain boundaries. We agree that further experiments on high strain deformation of fine-grained ice are needed to test the different hypothesis of the deformation mechanisms involved. We
40  have added the reference by Peternell et al. (2019) as this shows very well how the grain size and grain size distribution can vary with deformation conditions.

**Page 4 line 6:** I take exception to the statement "while the microstructures in the glacier ice suggest that GSS mechanisms may be important" This is not true and is not accepted by most members of the ice community.

There seems to be a major misunderstanding here. By Glacial age we mean, the ice that was deposited as snow in the last glacial maximum, which occurs in NEEM at depths of 1419 to 2207m, while the referee has apparently understood the entire 2000m column of ice. We have modified the text to avoid this misunderstanding. As already noted above, the sentence on page 4 line 5-6 in the revised manuscript does not dismiss any previous work and we definitely did not assume that GSS may dominate through the NEEM column. The phrase "while the microstructures in the glacier ice suggest that GSS mechanisms may be important" refers to the interpretation of the microstructures in the Glacial ice by Goldsby and Kohlstedt (2002), Faria et al (2014) and Saruya et al. (2019). To avoid any misunderstanding, we have expanded and rewritten this paragraph to include more studies on other Greenland ice cores as well as the NEEM papers that we mentioned.

**2.1 Study site:** There is overlap between Part 1 and the Part 2companion paper. It would be good if these could be combined and much of the unnecessary detail in Part 2 be removed. I also feel that **section 2.3** boundary conditions could be included in this section. A more concise description could be achieved by amalgamating these two parts.

We have retained the structure of two companion papers, particularly because the second referee as well as the editor clearly support this subdivision. Another argument is the high interest in the glaciological community on temperate deep ice, as in the deep ice flow descriptions are less well known than compared to cold ice. We have kept the section on boundary conditions separate from the section on the study site, because that part is concerned with the modelling rather than with observations from the NEEM ice core.

**2.2 Composite flow law**: Given my scepticism about the composite flow law this section is well argued. However, I would like to see more discussion about other flow laws e.g. Glens as this is what everything is being compared against and is better expressed in Part 2.
**Section 4 _ Results**: Initially I found this section difficult to read and verbose, but after reading **section 5.1** some clarification occurred (Except delete discussion about Olivine). It may be worthwhile to reword and shorten this section.

We did not delete the discussion about olivine because this is the one of the few studies where the effect of grain size distribution on GSS creep has been investigated, so it is highly relevant to compare the results from our study with other work on the same subject. Such model inter-comparison, as popular and needed for e.g. ice sheet modelling are also important in the development of micro-scale models for material properties.

**Section 5.3**: In reading this section and comparing figures 10a with 10b the significant differences, implications and the derivation of the stress exponent between these models are not clearly enunciated in the first part of this discussion. I believe with rearranging this section it will become clearer to the reader. Also could these two figures be combined as there is overlap?

We have changed the title of the section, as we investigated the effect of the assumed stress on the dominant deformation mechanisms, rather than looking at how the stress sensitivity to deformation varied with

depth. We did not combine figure 10a and 10b, because this makes the figure difficult to read, with many crossing lines.

**Section 5.5.**
This section in its present form should be deleted. Much of the discussion here is about other mineral systems and is irrelevant to the present models. There is substantial overlap with what is said in the following section. Only statements such as below should go into a reworded introduction.
"The c-axis eigenvalues show a minor variability in the glacial ice of the NEEM ice core (Eichler et al., 2013; Montagnat et al., 2014) where the layers of high strain rate are predicted. The strong development of CPO and the development of substructures with depth indicate that large amounts of strain are accommodated by basal slip of dislocations in the NEEM ice core."
Alternatively, if Parts 1 and 2 are combined into a single paper then the authors would address the concerns of Referee 1. The statement at the top of page 10 is vague

As noted above, we have rewritten section 5.5 substantially to account for these comments. We removed some references on other materials, but we kept the reference to olivine, because work on olivine shows that a strong CPO can develop during GSS creep in that material. Further experimental work is needed on ice to see if a CPO also develops in the GSS regimes found by Goldsby and Kohlstedt (2001) and Saruya et al. (2019).

**Section 5.6:** The overall statements in this section are almost contradictory with the last paragraph of this section really invalidating some of the reasons of applying these GBS-limited models.

We have merged the last two sections to account for this comment.

**The conclusions** to this paper rely solely on strain models and the effect of a temperature input is only mentioned briefly in results (page 10. Line 30); this is really left to Part 2. There is no discussion of possible limitations in the application of these models, except in **section 5.6** and this should have been better integrated with the conclusions. There should be a discussion and comparison to what is happening in the lower levels of the ice core but this is left to Part 2.
I would strongly recommend papers 1 and 2 are combined. A shorter more concise paper will have a far bigger impact. For instance, a good comparison between differences between Fig. 10 in Part 1 Vs. Fig. 4 in Part 2 would better explain the differences different models as they relate to temperature and strain regimes. It would have been good if the scales for Effective strain rates were the same.

In the revision of the paper we have emphasized the limitations of our approach and we have added statements to the abstract and conclusions to point out that while the models may be appropriate for the Glacial ice, the predictions of the models are inconsistent with observations in the Holocene ice. We conclude that the occurrence of dynamic recrystallization processes involving local strain induced grain boundary migration in natural ice is the probable reason why the models underestimate the importance of dislocation creep processes in the Holocene ice. As noted above we have not combined the papers. As the reviews of our part 1 and part 2 papers have shown, the topics of GSS creep and pre-melting in ice are controversial, so we think it is important to discuss the reasons for this controversy in detail and hopefully to remove some misunderstandings in the debate. While the detailed deformation mechanisms in fine-grained ice are controversial, there is a

consensus that the low stress deformation of ice, involves slip on the easy basal slip system, accommodated or enhanced in some way by grain boundary processes, resulting in a grain size sensitive strain rate (Goldsby 2006, Montagnant et al. 2003, Montagnant and Duval 2006, Saruya et al. 2019).

Response to referee report 2 by David Prior.

We thank the referee for the helpful comments. We have made the following changes exactly as suggested by the referee and we have also added the suggested references.

Line 10: Add a new sentence after "..experimental data": "This analysis highlighted an error in one part of the published flow law. The error was confirmed by others (Goldsby pers comm; Prior Pers comm) and is explained and corrected in the next two paragraphs". Then start a new paragraph with "A comparison…."

Line 21,22: Remove most of these two lines and start the paragraph with "We modified the flow law…."

20  Page 1. Line 15: comma after slip. Hyphen between rate and limited (and if you do hyphenate- do for whole manuscript).
Page 1. Line 16: comma after slip. Hyphen between rate and limited.
Page 3. Line 11: Replace "only one defm mech, that is" with "just".
Page 3. Line 13: Key reference for n=4 at Tertiary creep would be (Durham et al., 1983).
25  Page 3. Line 22: The new bit in red does not make that much sense. I think the alternative you are trying to suggest is a flow law that reflects directly the behavior at steady state or in Tertiary creep. Evidence that this has a different stress exponent in (Glen, 1953, 1955; Qi et al., 2017; Treverrow et al., 2012).
30  Page 9. Line 30: Remove "Therefore".
Page 14. Line 3: The low stress different n value is also shown and commented on in (Bons et al., 2018).
Page 17. Line 20: Remove the first sentence. Results using a flow law shown to be wrong are irrelevant.

We have made the following changes to take account of the comments.

Page 2. Line 5: Remove GBM from the list- it is not a deformation mechanism. After the citations on line 7 you could add "… and associated with recrystallization mechanisms, including GBM."

We left GBM in the list but changed the wording to refer to processes rather than mechanisms.

5 ….the internal deformation of the polycrystalline ice, which is governed by various processes like dislocation creep, grain boundary migration (GBM) and grain boundary sliding (GBS)….

Page 16. Line 7: I don't think you have spelt out "SIBM". Earlier in the manuscript (see page 2 comment) you use GBM. Maybe replace GBM with SIBM or better

10 with SIGBM. Just pick one to be consistent and then change all occurrences.

We kept SIBM and have used this throughout manuscript.

Page 17. Line 31. I think the term "margins" will be confusing here. You have used margins

15 earlier and so the context is set. Is it also worth mentioning other potentially high stress settings such as basal zones and lateral shear margins?

We changed the word margins to edges, but did not add basal zones and lateral shear margins because we are unsure if these are always high stress environments.

[revised manuscript text omitted]